# The indelible toll of enteric pathogens: Prevalence, clinical characterization, and seasonal trends in patients with acute community-acquired diarrhea in disenfranchised communities

**Marwan Osman**[1,2], **Issmat I. Kassem**[3], **Fouad Dabboussi**[4], **Kevin J. Cummings**[2], **Monzer Hamze**[4] *

**1** Cornell Atkinson Center for Sustainability, Cornell University, Ithaca, NY, United States of America, **2** Department of Public and Ecosystem Health, College of Veterinary Medicine, Cornell University, Ithaca, NY, United States of America, **3** Center for Food Safety and Department of Food Science and Technology, University of Georgia, Griffin, GA, United States of America, **4** Laboratoire Microbiologie Santé et Environnement (LMSE), Doctoral School of Sciences and Technology, Faculty of Public Health, Lebanese University, Tripoli, Lebanon

* mhamze@monzerhamze.com

## Abstract

### Background

There is little information on the epidemiology of enteric pathogens in Lebanon, a low- and middle-income country that suffers from a myriad of public health challenges. To address this knowledge gap, we aimed to assess the prevalence of enteric pathogens, identify risk factors and seasonal variations, and describe associations between pathogens among diarrheic patients in the Lebanese community.

### Methodology and principal findings

A multicenter cross-sectional community-based study was conducted in the north of Lebanon. Stool samples were collected from 360 outpatients suffering from acute diarrhea. Based on fecal examination using the BioFire® FilmArray® Gastrointestinal Panel assay, the overall prevalence of enteric infections was 86.1%. Enteroaggregative *Escherichia coli* (EAEC) was the most frequently identified (41.7%), followed by enteropathogenic *E. coli* (EPEC) (40.8%) and rotavirus A (27.5%). Notably, two cases of *Vibrio cholerae* were identified, while *Cryptosporidium* spp. (6.9%) was the most common parasitic agent. Overall, 27.7% (86/310) of the cases were single infections, and the majority, 73.3% (224/310), were mixed infections. Multivariable logistic regression models showed that enterotoxigenic *E. coli* (ETEC) and rotavirus A infections were significantly more likely to occur in the fall and winter compared to the summer. Rotavirus A infections significantly decreased with age but increased in patients living in rural areas or suffering from vomiting. We identified strong

**Data Availability Statement:** All code necessary to replicate the analysis is publicly available (DOI: 10.6084/m9.figshare.22013054).

**Funding:** This work was supported by a humanitarian donation from BioFire Diagnostics and Fondation Merieux USA. Marwan Osman is supported by the Atkinson Postdoctoral Fellowship (Cornell University). The donors and funders had no role in study design, data collection and analysis, decision to publish, or preparation of the manuscript.

**Competing interests:** The authors have declared that no competing interests exist.

associations in the co-occurrence of EAEC, EPEC, and ETEC infections and a higher percentage of rotavirus A and norovirus GI/GII infections among EAEC-positive cases.

## Conclusions

Several of the enteric pathogens reported in this study are not routinely tested in Lebanese clinical laboratories. However, anecdotal evidence suggests that diarrheal diseases are on the rise due to widespread pollution and the deterioration of the economy. Therefore, this study is of paramount importance to identify circulating etiologic agents and prioritize dwindling resources to control them and limit outbreaks in the future.

## 1. Introduction

Enteric infections represent a significant public health problem worldwide, greatly impacting human health and the economy [1, 2]. These pathogens spread commonly via the fecal-oral route following direct and/or indirect contact with the infectious agents which include human-to-human, zoonotic, waterborne, and foodborne transmission [3, 4]. The typical clinical features of these infections include gastrointestinal symptoms such as diarrhea, abdominal pain, vomiting, fever, nausea, malaise, and dehydration [5]. Gastrointestinal infections are widespread among children and constitute a neglected public health threat in underprivileged and disenfranchised populations, particularly in countries affected by adverse socioeconomic factors [6]. Indeed, diarrhea is the second leading cause of death in children younger than 5 years old, with an estimated 500,000 cases annually, many of which occur in low- and middle-income countries (LMICs) [7].

A broad range of enteric bacterial, viral, or parasitic pathogens can cause diarrhea and associated short- and long-term complications. Host vulnerabilities associated with poor nutritional status, anemia, inadequate sanitation and hygiene, overcrowded living conditions, and gaps in health literacy could facilitate repeated infections with certain pathogens, leading to severe consequences, including enteric and systemic inflammation, increased risk of stunting, impaired cognitive development, and/or death [2, 8–12]. Recent data from the Global Enteric Multicenter Study (GEMS) on the burden and etiology of diarrhea among children residing in developing countries showed that rotavirus, *Cryptosporidium* spp., enterotoxigenic *Escherichia coli* (ETEC), and *Shigella* spp. are the major causes of diarrheal diseases and death [6]. Furthermore, norovirus infections are the principal cause of foodborne disease outbreaks and are associated with approximately 20% of diarrhea cases, with similar prevalence in both children and adults [13, 14]. Hundreds of millions of people across the globe are estimated to have clinically diagnosable campylobacteriosis and giardiasis annually [15], while cholera continues in disaster- and war-inflicted countries such as Yemen [16], South Sudan [17], and recently Syria and Lebanon [18], with an estimated annual toll of 1.3–4.0 million cases globally [19]. Taken together, it is clear that gastrointestinal infections continue to contribute significantly to morbidity and mortality across the globe, with developing countries being the most affected.

Enteric pathogens can sometimes be detected in the stool of healthy asymptomatic individuals [20, 21]. This is also important, because asymptomatic carriage of enteric pathogens acts as a potential reservoir of infection and contributes to the high burden of gastrointestinal illnesses, promoting outbreaks and weakening the effectiveness of public health interventions [22, 23]. In addition, recent findings revealed that subclinical carriage of various enteric pathogens impacts

the growth of children [24]. Notably, *Shigella* spp., ETEC, *Campylobacter* spp., and *Cryptosporidium* spp. have been associated with gut inflammation and stunting [25–28].

In Lebanon, as in other LMICs, enteric infections remain anecdotally responsible for a significant degree of morbidity. However, the true prevalence and burden of enteric pathogens remain largely uncharacterized in Lebanon, largely due to 1) the lack of robust and sustainable epidemiological surveillance programs and 2) the inherent limitations of traditional diagnostic tools adopted by the majority of clinical laboratories that do not allow the detection of many pathogens. In Lebanon, the list of commonly targeted pathogens is usually limited to 1) rotavirus and adenovirus which are detected by a rapid qualitative immunochromatographic assay, 2) *Salmonella* spp. and *Shigella* spp. identified by conventional bacterial culture and biochemical analysis, and 3) a few enteric parasites (*Entamoeba histolytica/dispar*, *Entamoeba coli*, *Giardia duodenalis*, *Ascaris lumbricoides*, *Taenia* spp., and *Hymenolepis nana*) examined by direct-light microscopy. Additionally, the slide agglutination test, with commercial enteropathogenic *E. coli* (EPEC) antisera, is only done on *E. coli* isolates from individuals younger than 2 years old. Two previous studies conducted in South Lebanon based on the aforementioned diagnostic tools have not been able to identify enteric pathogens in approximately half of the hospitalized children with acute gastroenteritis [29, 30]. Another study in North Lebanon using stool culture and a commercial microarray assay (CLART® Enterobac) identified only 1.3% and 19% of enteric pathogens in children with no rotavirus or adenovirus infections, respectively [31]. In comparison, other reports based on molecular tools but targeting specific enteric pathogens, have shown a markedly different prevalence of pathogens. For example, 11% of hospitalized patients presenting with gastrointestinal disorders carried *Cryptosporidium* spp. [5]. *Campylobacter* spp. was detected in 11.1%-21.5% of diarrheic patients [32–34]. In children, 28.5% and 10.4% of stool samples were contaminated with *G. duodenalis* and *Cryptosporidium* spp. [4]. EPEC, *Shigella* spp., and *Salmonella* spp. have been found in 8.8%, 7.5%, and 2.5% of diarrheic children [31]. Rotavirus and norovirus were detected using a multiplex PCR in 48% and 6% of stool specimens collected from children hospitalized for severe acute gastroenteritis, respectively [35]. Taken together, current data on enteric pathogens circulation in Lebanese communities might be underestimated due to several interlinked factors related to the availability of resources and limitations in the investigation approach.

The use of robust molecular diagnostic methods that target multiple etiologic agents increases the sensitivity and specificity of detection for most enteric pathogens. To date, in-depth surveillance studies of enteric pathogens using these molecular diagnostic tools have been lacking in Lebanon. In addition, only very scant data are available on the potential risk factors and temporal trends associated with enteric pathogens in Lebanon. To address these gaps and better understand the epidemiology of enteric infections, we aimed to (i) determine the prevalence of enteric pathogens, (ii) identify risk factors and seasonal variations, and (iii) describe associations between pathogens among diarrheic patients in a Lebanese community that is at high risk of infection due to shortage in critical medical and nutritional necessities, weak water, sanitation and hygiene (WASH) programs, and a precipitous economic collapse [36–38]. Given the absence of routine and reliable diagnosis of enteric pathogens in Lebanon, this study provides crucial data that will assist stakeholders in prioritizing resources for beneficial interventions to control enteric infections in vulnerable populations.

## 2. Methods

### 2.1. Ethics approval

This study received the approval (CE-EDST-2-2020) of the ethical committee of the Doctoral School of Science and Technology at the Lebanese University (authorized by the Lebanese

Ministry of Public Health). Written informed consent was obtained from each patient or a legally authorized representative of the patient after a clear explanation of the research objectives. The data were analyzed anonymously.

## 2.2. Study design

This multicenter cross-sectional community-based study was conducted at two tertiary care facilities (El Youssef Hospital Center and Nini Hospital) that provide routine health and clinical laboratory services. The El Youssef Hospital Center and Nini Hospital are located in Halba and Tripoli, the capital cities of a rural (Akkar) and an urban (North) governorate in the north of Lebanon, respectively. The population eligible for the study consisted of any patient with acute diarrheal disorders attending one of the two healthcare facilities between July 2020 and July 2021. Participants met all of the following inclusion criteria: (i) age > 1-month-old, (ii) suffering from acute diarrheal disorders, (iii) onset of gastrointestinal-like symptoms within the last 14 days, and (iv) informed consent statement signed by the patient or a legally authorized representative. Exclusion criteria were i) any indication of healthcare-associated gastroenteritis (e.g., hospitalization for 2 or more days within 90 days of infection, residence in a nursing home or long-term care facility, recent antimicrobial therapy within 30 days prior to the current infection), chronic diarrhea (e.g., gastrointestinal-like symptoms lasting 14 days or longer), or any patient who had documented history of non-infectious etiology. Clinical microbiologists in the hospital evaluated each case based on the inclusion and exclusion criteria and the case report form. Each participant or legal representative completed a questionnaire addressing sociodemographic information (e.g., age, sex, residence, nationality, social status, educational level), behavioral habits (e.g., washing of fruits and vegetables, swimming, type of drinking water, frequency of eating outside home), presence of gastrointestinal symptoms (e.g., moderate-to-severe diarrhea (MSD), abdominal pain, vomiting, fever, and discomfort), and medical treatments. MSD was defined as an acute episode of diarrhea with more than three watery stools per day that started within the previous week [39]. A minimum sample size of 246 patients was needed to estimate the prevalence of enteric pathogens, their association with demographic characteristics, and potential risk factors with a confidence level of 95% and a margin of error of 5% (based on population size [1,000,000 individuals] and an estimated prevalence of enteric pathogens [80%]). The calculation of the sample size was done using Epi info Software (v7.2.5) (https://www.cdc.gov/epiinfo/pc.html). However, we assessed a total number of 360 patients for this study to sufficiently represent the populations of Akkar (population n = 423,596) and North (n = 807,204) governorates, which comprise ~ 1/6th of the Lebanese population.

## 2.3. Microbiologic analysis of biological samples

One fresh stool sample (>10g) per individual was collected in a sterile container, kept at 4˚C, and rapidly transported to the Laboratoire Microbiologie Santé et Environnement (LMSE) in Tripoli, Lebanon. All stool samples were analyzed using the BioFire® FilmArray® Gastrointestinal Panel assay (bioMérieux) that is designed to detect enteric pathogens [enteroaggregative *E. coli* (EAEC), EPEC, ETEC *lt* and *st*, Shiga toxin-producing *E. coli* (STEC) stx1 and stx2 (including specific detection of *E. coli* O157), enteroinvasive *E. coli* (EIEC)/*Shigella* spp., *Campylobacter* spp., *Salmonella* spp., *Clostridium difficile* toxin A/B, *Plesiomonas shigelloides*, *Yersinia enterocolitica*, *Vibrio* spp. (including specific detection of *V. cholerae*), adenovirus F 40/41, astrovirus, norovirus GI/GII, rotavirus A, sapovirus (I, II, IV, and V), *Cryptosporidium* spp., *G. duodenalis*, *Cyclospora cayetanensis*, and *Entamoeba histolytica*] [40, 41]. The assay was performed following the manufacturer's instructions and controls and using good clinical

laboratory practices that included in-house standard operating procedures defined for this study. No conventional microbiological assays (e.g., culture, parasitological analyses) were carried out. Positive and negative results were promptly reported to patients and their respective clinicians to promote the selection of appropriate treatment and prevent the misuse of antimicrobials.

## 2.4. Statistical analysis

Data recorded in the questionnaire and the laboratory results were checked for completeness and consistency before analysis. The statistical analysis was carried out using the R software (R Core team, version 4.1.0; R Studio, version 2022.07.1–554). The dataset was imported for cleaning, variable coding, and analysis. Descriptive analysis was performed on all variables using several packages (e.g., dplyr, stringr, prettyR, summarytools) and results were illustrated using the ggplot2 R package. All code necessary to replicate the analysis is publicly available (DOI: 10.6084/m9.figshare.22013054). Data were presented as mean [min-max] for continuous variables and as frequency distributions for categorical variables. To predict the determinants of enteric infections at the univariate level, we compared the differences across groups using the t-test and the Pearson chi-squared test for continuous and categorical variables, respectively. The presence of an enteric pathogen (yes/no) was the outcome and sociodemographic and potential risk factors were the explanatory variables. Multivariable logistic regression models were created to test for the determinants of the occurrence of (i) enteric infections and (ii) each pathogen versus none. Moreover, we predicted the associations between enteric pathogens using a third group of multivariable logistic regression models. Infection with select enteric pathogens (prevalence>5%) was considered the explanatory variable. Backward elimination was performed to confirm the determinants that best predicted the occurrence of enteric infections and the associations between pathogens. Results from univariable and multivariable logistic regression analysis were expressed as odds ratio (OR) with 95% confidence intervals (CI). All statistical tests were two-sided, with a type I error set at $\alpha = 0.05$. Since many regression models were created (**Table 6**), we performed the Benjamini-Hochberg method to adjust the calculated P-values to control the false discovery rate to 0.05 [42].

## 3. Results

A total of 360 patients suffering from acute community-acquired diarrhea (201 males, 159 females) were enrolled in this study. The majority of patients (274) lived in rural areas (mostly patients from El Youssef Hospital Center), while the remaining 66 lived in urban areas (mostly patients from Nini Hospital). The patients were between 1 and 91 years old (mean age: $8.3 \pm 17.1$ years old) and were divided into three groups; preschoolers ($\leq 5$ years, 78.9%), children (6–17 years, 13.9%), and adults ($\geq 18$ years, 7.2%). Gastrointestinal symptoms varied among the patients. MSD, abdominal pain, vomiting, and fever were reported in 87.5% (300/343), 34.7% (119/343), 57.4% (197/343), and 61.5% (211/343) of patients, respectively (**Table 1**). Although we checked the data for completeness and consistency, we failed to get the information about the gastrointestinal symptoms of 17 participants in the study. The mean number of admitted patients was $27.7 \pm 10.6$ per month, with a maximum number of samples collected between September and November and a minimum number of samples collected during the first and last months of the study (**Fig 1**).

Overall, 86.1% (310/360) of the patients were found to be positive for at least one enteric pathogen using the BioFire® FilmArray® Gastrointestinal Panel assay. Bacterial infections were most common among patients (76.1%, 274/360), followed by viral (42.8%, 154/360) and parasitic (10.3%, 37/360) infections. Almost 70% of the individuals carried at least one

**Table 1. Sociodemographic characteristics of the study population and the prevalence of enteric pathogens.**

| | Total | | Missing information | |
|---|---|---|---|---|
| | **N** | **%** | **n** | **%** |
| **Hospital** | | | 0 | 0 |
| El Youssef Hospital | **266** | 78.9 | | |
| Nini Hospital | **94** | 26.1 | | |
| **Age (mean [min-max])** | 8.3 [1–91 years] | | 0 | 0 |
| **Age class** | | | 0 | 0 |
| ≤5 years | **284** | 78.9 | | |
| 6–17 years | **50** | 13.9 | | |
| ≥18 years | **26** | 7.2 | | |
| **Sex** | | | 0 | 0 |
| Female | **159** | 44.2 | | |
| Male | **201** | 55.8 | | |
| **Region** | | | 20 | 5.6 |
| Urban | **66** | 19.4 | | |
| Rural | **274** | 80.6 | | |
| **Wash fruits and vegetables before consumption** | | | 20 | 5.6 |
| Yes | **317** | 93.2 | | |
| No | **23** | 6.8 | | |
| **Using antiseptic products in washing** | | | 20 | 5.6 |
| Yes | **54** | 17.0 | | |
| No | **263** | 83.0 | | |
| **Swim (during the last 14 days)** | | | 27 | 7.5 |
| Yes | **36** | 10.8 | | |
| No | **297** | 89.2 | | |
| **Eating outside household** | | | 32 | 8.9 |
| 0-1/week | **266** | 81.1 | | |
| >1/week | **62** | 18.9 | | |
| **Drinking water** | | | 40 | 11.1 |
| Treated | **202** | 63.1 | | |
| Untreated | **118** | 36.9 | | |
| **Season** | | | 0 | 0 |
| Summer | **93** | 25.8 | | |
| Fall | **108** | 30.0 | | |
| Winter | **88** | 24.5 | | |
| Spring | **71** | 19.7 | | |
| **Detection of enteric pathogens** | | | 0 | 0 |
| Yes | **310** | 86.1 | | |
| No | **50** | 13.9 | | |
| ***Bacteria*** | *274* | *76.1* | 0 | 0 |
| ***Diarrheagenic Escherichia coli / Shigella* spp.** | *247* | *68.6* | | |
| EPEC | **147** | 40.8 | | |
| EAEC | **150** | 41.7 | | |
| ETEC *lt/st* | **71** | 19.7 | | |
| EIEC/*Shigella* spp. | **68** | 18.9 | | |
| STEC[¶] *stx1/stx2* | **16** | 4.4 | | |
| ***Other pathogenic bacteria*** | *107* | *29.7* | | |
| *Campylobacter* spp. | **52** | 14.4 | | |

(*Continued*)

**Table 1.** (Continued)

| | Total | | Missing information | |
|---|---|---|---|---|
| | **N** | **%** | **n** | **%** |
| *Salmonella* spp. | **34** | 9.4 | | |
| *Clostridium difficile* toxin A/B | **21** | 5.8 | | |
| *Plesiomonas shigelloides* | **7** | 1.9 | | |
| *Vibrio cholerae* | **2** | 0.6 | | |
| *Yersinia enterocolitica* | **0** | 0 | | |
| **Virus** | **154** | **42.8** | 0 | 0 |
| Rotavirus A | **99** | 27.5 | | |
| Norovirus GI/GII | **27** | 7.5 | | |
| Adenovirus F40/41 | **18** | 5.0 | | |
| Sapovirus (I, II, IV, and V) | **17** | 4.7 | | |
| Astrovirus | **6** | 1.7 | | |
| **Parasite** | **37** | **10.3** | 0 | 0 |
| *Cryptosporidium* spp. | **25** | 6.9 | | |
| *Giardia duodenalis* | **13** | 3.6 | | |
| *Cyclospora cayetanensis* | **0** | 0 | | |
| *Entamoeba histolytica* | **0** | 0 | | |
| **Digestive symptoms** | | | 17 | 4.7 |
| Moderate-to-severe diarrhea | **300** | 87.5 | | |
| Abdominal pain | **119** | 34.7 | | |
| Vomiting | **197** | 57.4 | | |
| Fever | **211** | 61.5 | | |

Missing information has been omitted. Data are presented as mean [min-max] for the continuous variable age and as frequency and percentage for categorical variables.

**ᶲ**Three STEC cases have been identified as *E. coli* O157

diarrheagenic *E. coli*, mainly enteroaggregative *E. coli* (41.7%) and EPEC (40.8%). *Campylobacter* spp. and *Salmonella* spp. were detected in 14.4% and 9.4% of the samples, respectively. Regarding viruses, rotavirus A (27.5%) and norovirus GI/GII (7.5%) were the primary viral causes of diarrhea. *Cryptosporidium* spp. (6.9%) was the most common parasitic agent, followed by *G. duodenalis* (3.6%) (**Fig 2**). No cases of *Yersinia enterocolitica*, *Entamoeba histolytica*, or *Cyclospora cayetanensis* were found. Two cases of *Vibrio cholerae* were identified in patients during the September-October period: a Lebanese male child (1 year) and a Lebanese female adult (29 years) living in a rural and an urban area, respectively. Surprisingly, in terms of patient behavior, washing fruits and vegetables before consumption was associated with enteric infections ($P < 0.001$).

Out of 310 infected patients, the distribution of enteric pathogens was 54.5% (169/310) in males and 45.5% (141/310) in females. When gastrointestinal symptoms were considered, most infections were associated with MSD (90.1%), followed by fever (65.3%), vomiting (61.2%), and abdominal pain (34.4%). Multivariable logistic regression analysis (**Table 2**) confirmed that vomiting occurred significantly more likely among infected cases [odds ratio (OR) = 2.34; 95% confidence interval (95% CI) = 1.10 to 5.05; $P = 0.03$] in comparison to non-infected cases. At the bivariate level, MSD and fever were also more common among infected cases compared to non-infected patients ($P < 0.05$).

Overall, 27.7% (86/310) of the cases constituted single infections and the remaining majority, 73.3% (224/310), were mixed infections (**Fig 3A**). Single infections mainly comprised

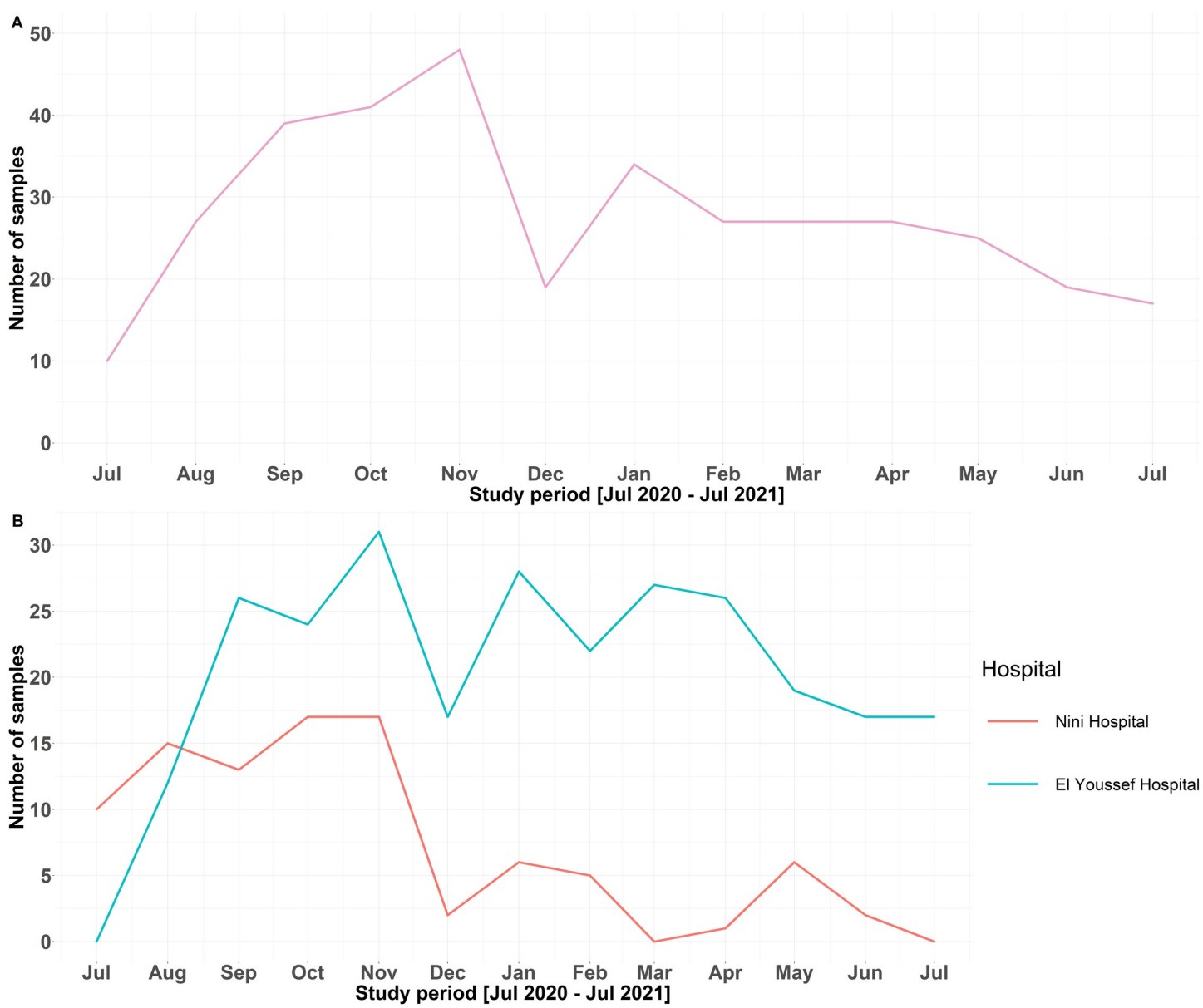

**Fig 1.** Total number of collected stool samples during the study period (A), stratified by health care settings (B).

rotavirus A (24.4%, 21/86), followed by EPEC (14%), *Salmonella* spp. (9.3%), EIEC (9.3%), EAEC (8.1%), and *Campylobacter* spp. (7%), *Cryptosporidium* spp. (5.8%), and *C. difficile* (5.8%) (**Fig 3B**). Mixed infections with two and three enteric pathogens were found in 25.5% and 24.2% of patients, respectively (S1 Fig). The most common mixed infection was with rotavirus A, EPEC, and EAEC (6.3%, 14/224), followed by rotavirus A with either EAEC (4.9%) or EPEC (4%). Three cases were coinfected with six pathogens, including *Campylobacter* spp. and EIEC/*Shigella* spp. which were present in all three cases.

The number of enteric pathogens was investigated during different months of the year to describe temporal trends (**Fig 4**). We found that the seasonal prevalence of ETEC and EIEC/*Shigella* spp. was higher during the summer and fall seasons compared to the winter and spring seasons (**Fig 4A**). Rotavirus A had a long peak that occurred between January and April, and

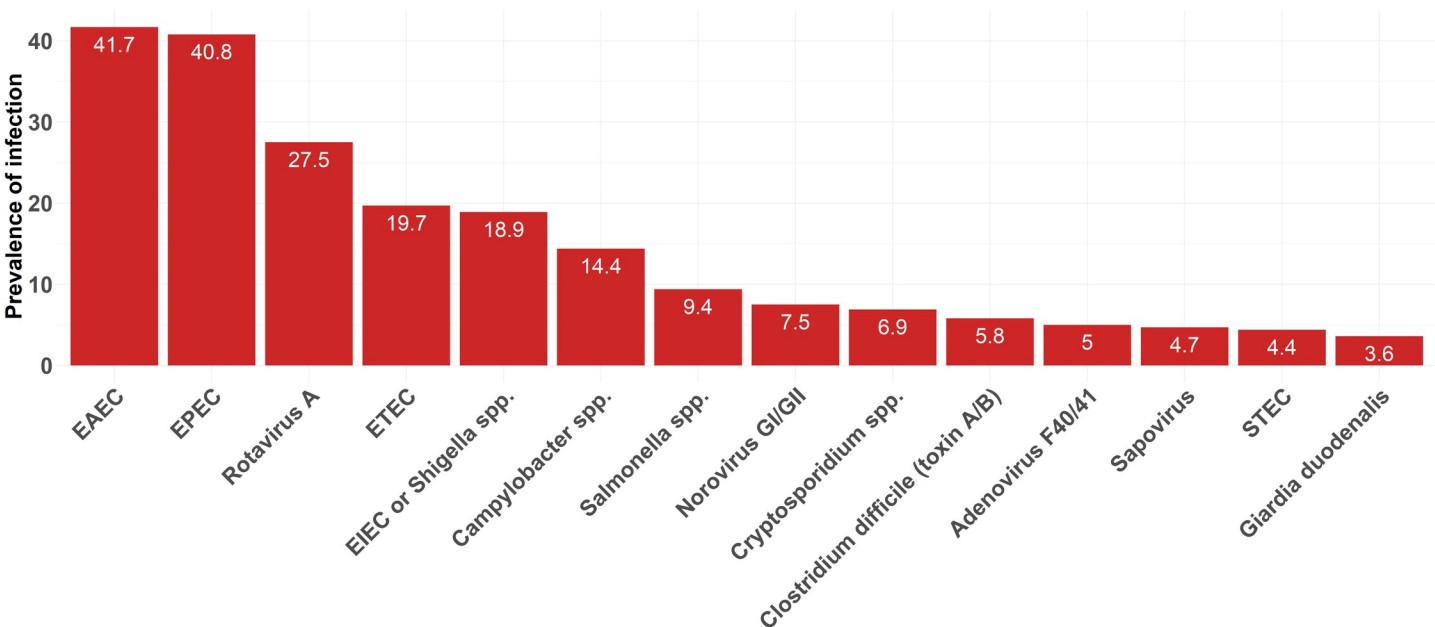

**Fig 2. Prevalence of enteric pathogens among patients suffering from acute community-acquired diarrhea in two Lebanese tertiary healthcare settings.** Enteric pathogens with a prevalence lower than 2% were not shown in this figure (*Plesiomonas shigelloides*: 1.9%, *Vibrio cholerae*: 0.6%, and astrovirus:1.7%).

norovirus GI/GII showed peaks in June and July (**Fig 4E**). The peak in cryptosporidiosis occurred in October (**Fig 4G**). When controlling for age, sex, living area, behavioral habits, season, and presence of gastrointestinal symptoms, the multivariable logistic regression analysis corroborated the descriptive analysis and showed that ETEC and EIEC/*Shigella* spp. were detected significantly less frequently during the winter and spring seasons in comparison to the summer. In addition, ETEC infections occurred significantly more commonly in the fall (OR = 2.22; 95% CI = 1.12 to 4.49; P = 0.02) in comparison to the summer (**Tables 3** and **4**). Identification of rotavirus A infections increased significantly during the winter months (OR = 5.15; 95% CI = 2.21 to 12.7; P < 0.001) in comparison to the summer (**Table 5**). Multivariable logistic regression models also identified that female patients were significantly more likely to be infected with ETEC (P = 0.002), while norovirus GI/GII infections were more frequently observed among those who swam during the previous 14 days (OR = 3.33; 95% CI = 1.12 to 8.85; P = 0.02). Rotavirus A infections decreased with age (OR = 0.68; 95% CI = 0.50 to 0.88; P = 0.01) but increased among patients living in rural areas (OR = 8.51; 95% CI = 1.51 to 162; P = 0.04) or those who suffered from vomiting (OR = 3.12; 95% CI = 1.55 to 6.56; P = 0.002). STEC-positive cases were significantly more likely to be associated with abdominal pain (OR = 4.67; 95% CI = 1.44 to 17.9; P = 0.01) after controlling for confounding factors.

Multivariable logistic regression analysis revealed several associations between the enteric pathogens. After controlling for other pathogens, we found strong relationships between EPEC, EAEC, and ETEC infections (OR > 2; P < 0.05), broadly capturing the mixed infection patterns that were mentioned earlier (**Fig 3B**). Furthermore, ETEC-positive cases showed a higher percentage of EIEC/*Shigella* spp. infections (and vice versa) (OR = 2.49; 95% CI = 1.33 to 4.60; P = 0.026). Rotavirus A was associated with an increase in the probability of EAEC infections (OR = 2.33; 95% CI = 1.39 to 3.91; P = 0.018) (and vice versa). A similar association was observed in the MLR model predicting EAEC infections in the norovirus GI/GII-positive cases (OR = 3.55; 95% CI = 1.50 to 8.94; P = 0.027) (and vice versa). However, rotavirus A was

**Table 2. Determinants of enteric infections among patients suffering from acute community-acquired diarrhea in two Lebanese tertiary health care settings according to sociodemographic and potential risk factors.**

| | Univariate analysis | | Model 1[i] | | | Model 2[ii] | | |
| --- | --- | --- | --- | --- | --- | --- | --- | --- |
| | Infection | | | | | | | |
| | % | P | adj. OR | 95%CI | P | adj. OR | 95%CI | P |
| **Age (continuous variable)** | | **0.04** | 0.99 | 0.97–1.01 | 0.21 | | | |
| **Sex** | | | | | | | | |
| Female | 88.7 | 0.27 | | | | | | |
| Male | 84.1 | | | | | | | |
| **Region** | | | | | | | | |
| Urban | 84.8 | 0.94 | | | | | | |
| Rural | 86.1 | | | | | | | |
| **Wash fruits and vegetables before consumption** | | | | | | | | |
| Yes | **88.3** | **<0.001** | 2.74 | 0.86–8.11 | 0.07 | 2.52 | 0.81–7.34 | 0.10 |
| No[1] | **60.9** | | | | | | | |
| **Using antiseptic products in washing** | | | | | | | | |
| Yes[1] | 81.5 | 0.14 | | | | | | |
| No | 89.7 | | | | | | | |
| **Swim (during the last 14 days)** | | | | | | | | |
| Yes | 97.2 | 0.09 | 4.60 | 0.84–86.6 | 0.15 | 4.86 | 0.90–90.8 | 0.14 |
| No[1] | 85.5 | | | | | | | |
| **Eating outside household** | | | | | | | | |
| 0-1/week | 86.8 | 0.85 | | | | | | |
| >1/week | 88.7 | | | | | | | |
| **Drinking water** | | | | | | | | |
| Treated | 89.1 | 0.34 | | | | | | |
| Untreated | 84.7 | | | | | | | |
| **Season** | | | | | | | | |
| Summer[1] | 88.2 | 0.17 | | | | | | |
| Fall | 90.7 | | 1.73 | 0.60–5.24 | 0.31 | 1.88 | 0.66–5.63 | 0.24 |
| Winter | 80.7 | | 0.56 | 0.21–1.44 | 0.24 | 0.61 | 0.23–1.52 | 0.29 |
| Spring | 83.1 | | 0.45 | 0.15–1.26 | 0.13 | 0.50 | 0.18–1.37 | 0.18 |
| **Moderate-to-severe diarrhea** | | | | | | | | |
| Yes | **88.3** | **<0.001** | 1.65 | 0.63–4.08 | 0.29 | | | |
| No[1] | **67.4** | | | | | | | |
| **Abdominal pain** | | | | | | | | |
| Yes | 84.9 | 0.87 | | | | | | |
| No | 86.2 | | | | | | | |
| **Vomiting** | | | | | | | | |
| Yes | **91.4** | **<0.001** | 2.02 | 0.92–4.50 | 0.08 | **2.34** | **1.10–5.05** | **0.03** |
| No[1] | **78.1** | | | | | | | |
| **Fever** | | | | | | | | |
| Yes | **91.0** | **<0.001** | 1.61 | 0.74–3.46 | 0.22 | 1.85 | 0.89–3.85 | 0.10 |
| No[1] | **77.3** | | | | | | | |

*Determinants of infection were predicted using univariate and multivariable analysis.

[i]The variables tested by univariate analysis that had a P value < 0.20 were included in Model 1 (multivariable logistic regression analysis).

[ii]In Model 2, a backward logistic regression model was created including only complete cases.

[1]Reference group.

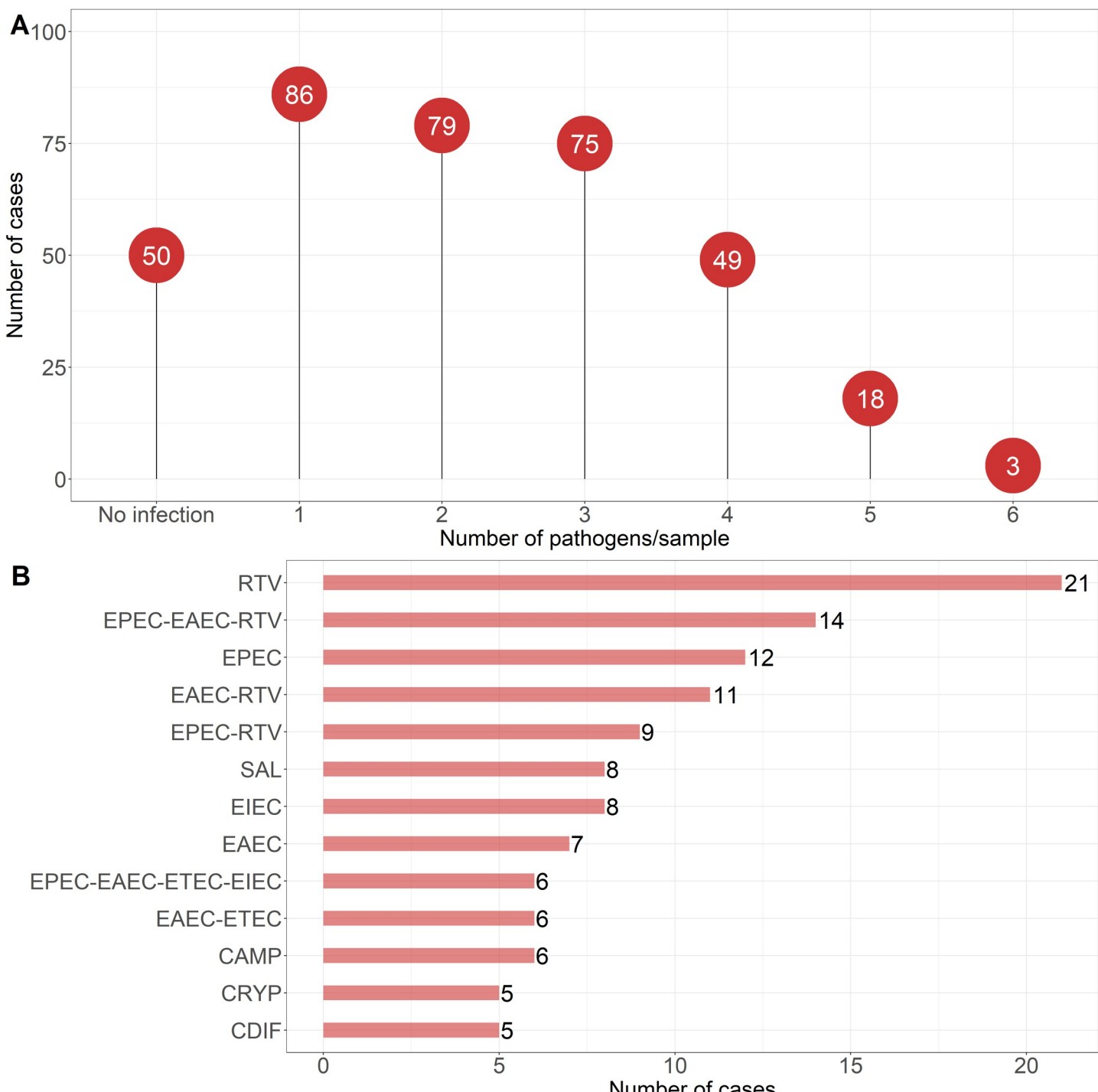

**Fig 3. Distribution of single and mixed enteric infections and most common pathogen association patterns among patients suffering from acute community-acquired diarrhea in two Lebanese tertiary healthcare settings.** The number of single, double, triple, quadruple, quintuple, and sextuple infections is shown (A). Only pathogen association patterns with ≥5 cases are shown (B). *CAMP: *Campylobacter* spp.; CDIF: *Clostridium difficile*; CRYP: *Cryptosporidium* spp.; EAEC: Enteroaggregative *Escherichia coli*; EIEC: Enteroinvasive *E. coli/Shigella* spp.; EPEC: Enteropathogenic *E. coli*; ETEC: Enterotoxigenic *E. coli*; RTV: Rotavirus A; SAL: *Salmonella* spp.

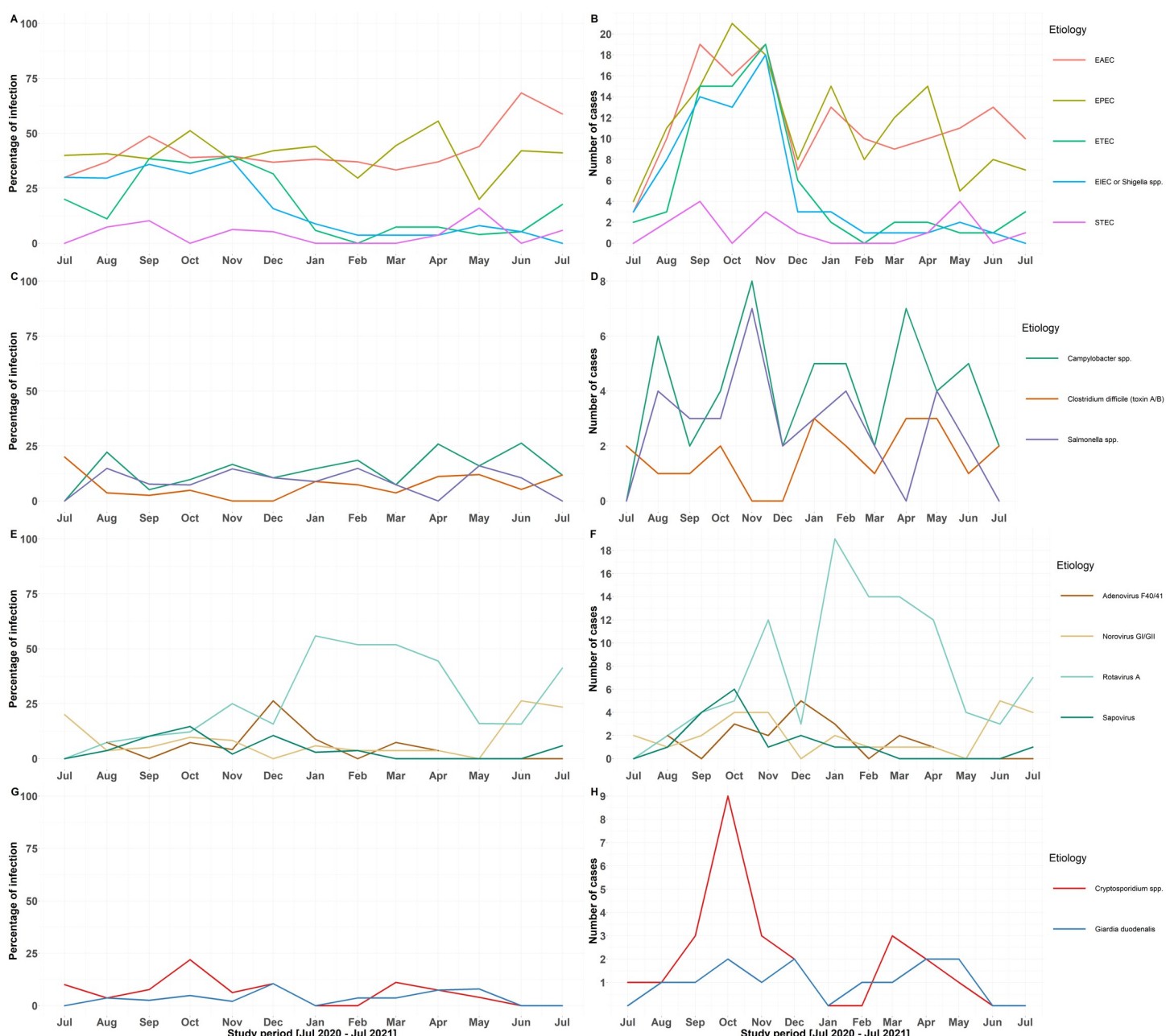

**Fig 4. Temporal trends in the prevalence and number of isolates of enteric pathogens among patients suffering from acute community-acquired diarrhea in two Lebanese tertiary healthcare settings during the study period.** (A and B) Diarrheagenic *Escherichia coli* / *Shigella* spp.; (C and D) Other pathogenic bacteria; (E and F) Viruses; and (G and H) Parasites.

found to be associated with a decrease in the probability of infection with ETEC (OR = 0.31; 95% CI = 0.14 to 0.64; P = 0.021), EIEC/*Shigella* spp. (OR = 0.32; 95% CI = 0.14 to 0.68; P = 0.027), and *Salmonella* spp. (OR = 0.22; 95% CI = 0.06 to 0.60; P = 0.034) (**Table 6**).

## 4. Discussion

This study generated unprecedented data on the etiology of diarrhea in the Lebanese community, which has been facing calamitous challenges, including COVID-19, shortages in critical

**Table 3. Determinants of enteric pathogens among patients suffering from acute community-acquired diarrhea using univariate analysis in two Lebanese tertiary health care settings according to sociodemographic characteristics and potential risk factors.**

| | Diarrheagenic Escherichia coli / Shigella spp. | | | | | | | | | | Other pathogenic bacteria | | | | | | Viruses | | | | | | | | Parasites | | | |
|---|---|---|---|---|---|---|---|---|---|---|---|---|---|---|---|---|---|---|---|---|---|---|---|---|---|---|---|---|
| | EP EC | | EA EC | | ET EC | | EIEC/Shigella spp. | | ST EC | | Campylobacter spp. | | Salmonella spp. | | Clostridium difficile | | Rotavirus A | | Norovirus GI/GII | | Adenovirus | | Sapovirus | | Cryptosporidium spp. | | Giardia duodenalis | |
| | % | P | % | P | % | P | % | P | % | P | % | P | % | P | % | P | % | P | % | P | % | P | % | P | % | P | % | P |
| **Age (continuous variable)** | | 0.87 | | 0.09 | | 0.08 | | 0.93 | | 0.41 | | 0.51 | | 0.56 | | 0.79 | | **<0.001** | | 0.07 | | **<0.001** | | 0.34 | | **0.002** | | 0.83 |
| **Sex** | | | | | | | | | | | | | | | | | | | | | | | | | | | | |
| Female | 45.3 | 0.16 | 43.4 | 0.63 | 25.8 | **0.01** | 21.4 | 0.35 | 3.8 | 0.77 | 13.8 | 0.89 | 7.6 | 0.36 | 6.3 | 0.92 | 30.2 | 0.37 | 5.7 | 0.33 | 5.0 | 1.0 | 4.4 | 1.0 | 7.6 | 0.85 | 1.9 | 0.20 |
| Male | 37.3 | | 40.3 | | 14.9 | | 16.9 | | 5.0 | | 14.9 | | 11.0 | | 5.5 | | 25.4 | | 9.0 | | 5.0 | | 5.0 | | 6.5 | | 5.0 | |
| **Region** | | | | | | | | | | | | | | | | | | | | | | | | | | | | |
| Urban | 42.4 | 1.0 | 33.3 | 0.12 | 27.3 | 0.09 | 24.2 | 0.31 | 4.6 | 0.90 | 7.6 | 0.10 | 15.2 | 0.12 | 4.6 | 1.0 | 1.5 | **<0.001** | 7.6 | 1.0 | 4.6 | 1.0 | 3.0 | 0.69 | 6.1 | 1.0 | 6.6 | 0.49 |
| Rural | 42.0 | | 44.9 | | 17.2 | | 17.9 | | 3.3 | | 16.1 | | 7.7 | | 5.5 | | 35.0 | | 7.7 | | 5.1 | | 5.1 | | 6.6 | | 3.3 | |
| **Wash fruits and vegetables before consumption** | | | | | | | | | | | | | | | | | | | | | | | | | | | | |
| Yes | 43.2 | 0.33 | 43.9 | 0.30 | 20.5 | 0.11 | 19.6 | 0.62 | 3.8 | 1.0 | 15.5 | 0.80 | 9.2 | 0.14 | 5.1 | 0.79 | 29.7 | 0.14 | 7.6 | 1.0 | 5.4 | 0.52 | 5.1 | 0.55 | 6.9 | 0.39 | 4.1 | 0.67 |
| No | 30.4 | | 30.4 | | 4.4 | | 13.0 | | 4.4 | | 0 | | 13.0 | | 8.7 | | 13.0 | | 8.7 | | 0 | | 0 | | 0 | | 0 | |
| **Using antiseptic products in washing** | | | | | | | | | | | | | | | | | | | | | | | | | | | | |
| Yes | 38.9 | 0.42 | 43.6 | 0.79 | 18.1 | 0.35 | 16.7 | 0.69 | 1.9 | 0.67 | 7.4 | 0.22 | 13.0 | 0.42 | 3.7 | 0.73 | 25 | 0.73 | 7.4 | 1.0 | 3.7 | 0.37 | 5.6 | 1.0 | 1.9 | 0.19 | 5.6 | 0.83 |
| No | 44.1 | | 45.8 | | 24.2 | | 20.2 | | 4.2 | | 17.1 | | 8.4 | | 5.3 | | 29.3 | | 7.6 | | 5.7 | | 4.9 | | 8.0 | | 3.8 | |
| **Swim (during the last 14 days)** | | | | | | | | | | | | | | | | | | | | | | | | | | | | |
| Yes | 50 | 0.42 | 52.8 | 0.30 | 25 | 0.51 | 33.3 | **0.04** | 7.6 | **0.02** | 11.1 | 0.69 | 16.7 | 0.38 | 2.8 | 0.20 | 16.7 | **0.03** | 16.7 | **0.04** | 2.8 | 0.79 | 13.9 | **0.02** | 8.3 | 0.93 | 5.6 | 0.93 |
| No | 41.4 | | 42.1 | | 18.9 | | 17.5 | | 1.8 | | 15.2 | | 8.8 | | 5.7 | | 32.3 | | 6.1 | | 5.4 | | 3.7 | | 6.4 | | 3.7 | |
| **Eating outside household** | | | | | | | | | | | | | | | | | | | | | | | | | | | | |
| 0–1/week | 42.1 | 0.95 | 42.5 | 0.64 | 18.1 | 0.35 | 18.4 | 0.78 | 4.1 | 1.0 | 13.9 | 0.38 | 8.3 | 0.20 | 5.3 | 0.95 | 33.1 | **0.003** | 7.9 | 0.31 | 3.7 | 0.37 | 3.8 | 0.11 | 6.8 | 1.0 | 3.0 | 0.14 |
| >1/week | 43.6 | | 46.8 | | 24.2 | | 21.0 | | 3.2 | | 19.4 | | 14.5 | | 6.5 | | 12.9 | | 3.2 | | 5.7 | | 9.7 | | 6.5 | | 8.1 | |
| **Drinking water** | | | | | | | | | | | | | | | | | | | | | | | | | | | | |
| Treated | 48.0 | 0.08 | 43.6 | 0.79 | 23.3 | 0.12 | 18.3 | 0.77 | 5.0 | 0.45 | 14.9 | 1.0 | 10.9 | 1.0 | 5.9 | 0.46 | 28.2 | 0.88 | 6.9 | 0.42 | 5.5 | 1.0 | 5.9 | 0.46 | 5.9 | 1.0 | 3.5 | 0.96 |
| Untreated | 37.3 | | 45.8 | | 15.3 | | 20.3 | | 2.5 | | 15.3 | | 7.6 | | 3.4 | | 29.7 | | 10.2 | | 5.1 | | 3.4 | | 5.9 | | 4.2 | |
| **Season** | | | | | | | | | | | | | | | | | | | | | | | | | | | | |
| Summer | 39.8 | 0.93 | 45.2 | 0.40 | 24.7 | **<0.001** | 26.9 | **<0.001** | 7.5 | 0.06 | 10.8 | 0.17 | 7.5 | 0.40 | 6.5 | 0.83 | 14.0 | **0.003** | 9.7 | 0.61 | 2.2 | 0.05 | 6.5 | **0.04** | 5.4 | **0.04** | 2.2 | 0.54 |
| Fall | 43.5 | | 38.9 | | 37.0 | | 31.5 | | 3.7 | | 13.0 | | 11.1 | | 1.9 | | 18.5 | | 7.4 | | 9.3 | | 8.3 | | 13.0 | | 4.6 | |
| Winter | 39.8 | | 36.4 | | 4.6 | | 5.7 | | 0 | | 13.6 | | 10.2 | | 6.8 | | 52.3 | | 4.6 | | 5.7 | | 2.3 | | 3.4 | | 2.3 | |
| Spring | 39.4 | | 47.9 | | 5.6 | | 5.6 | | 7.0 | | 22.5 | | 8.5 | | 9.9 | | 26.8 | | 8.5 | | 1.4 | | 0 | | 4.2 | | 5.6 | |
| **Moderate-to-severe diarrhea** | | | | | | | | | | | | | | | | | | | | | | | | | | | | |
| Yes | 42.0 | 1.0 | 43.7 | 0.36 | 20.3 | 0.25 | 18.3 | 0.57 | 3.7 | 1.0 | 14.7 | 0.76 | 10.0 | 0.40 | 5.7 | 0.58 | 29.0 | 0.55 | 7.3 | 0.88 | 5.0 | 1.0 | 5.0 | 0.70 | 6.0 | 0.62 | 3.7 | 1.0 |
| No | 41.9 | | 34.9 | | 11.6 | | 23.3 | | 4.7 | | 11.6 | | 4.7 | | 2.3 | | 23.3 | | 9.3 | | 4.7 | | 2.3 | | 9.3 | | 4.7 | |
| **Abdominal pain** | | | | | | | | | | | | | | | | | | | | | | | | | | | | |
| Yes | 38.9 | 0.43 | 37.0 | 0.21 | 15.1 | 0.21 | 14.3 | 0.14 | 3.8 | 1.0 | 18.5 | 0.14 | 8.4 | 0.81 | 5.0 | 1.0 | 31.1 | 0.47 | 6.7 | 0.82 | 4.2 | 0.84 | 1.7 | 0.11 | 5.9 | 0.95 | 3.4 | 1.0 |
| No | 43.8 | | 45.5 | | 21.4 | | 21.4 | | 3.8 | | 12.1 | | 9.8 | | 5.4 | | 26.8 | | 8.0 | | 5.4 | | 6.3 | | 6.7 | | 4.0 | |
| **Vomiting** | | | | | | | | | | | | | | | | | | | | | | | | | | | | |
| Yes | 44.2 | 0.40 | 46.2 | 0.14 | 18.8 | 0.98 | 19.3 | 0.96 | 4.1 | 0.98 | 15.2 | 0.67 | 8.1 | 0.48 | 5.6 | 0.94 | 40.1 | **<0.001** | 6.6 | 0.55 | 4.1 | 0.52 | 4.1 | 0.72 | 6.6 | 1.0 | 3.1 | 0.58 |
| No | 39.0 | | 37.7 | | 19.9 | | 18.5 | | 3.4 | | 13.0 | | 11.0 | | 4.8 | | 12.3 | | 8.9 | | 6.2 | | 5.5 | | 6.2 | | 4.8 | |
| **Fever** | | | | | | | | | | | | | | | | | | | | | | | | | | | | |
| Yes | 44.1 | 0.38 | 46.0 | 0.13 | 19.0 | 0.98 | 20.4 | 0.48 | 3.8 | 1.0 | 15.6 | 0.45 | 10.9 | 0.28 | 6.2 | 0.48 | 33.7 | **0.01** | 5.7 | 0.14 | 5.2 | 0.98 | 3.8 | 0.48 | 6.2 | 0.99 | **1.9** | **0.04** |
| No | 38.6 | | 37.1 | | 19.7 | | 16.7 | | 3.8 | | 12.1 | | 6.8 | | 3.8 | | 19.7 | | 10.6 | | 4.6 | | 6.1 | | 6.8 | | 6.82 | |

*Determinants of infection were predicted using univariate analysis.

**Table 4. Determinants of bacterial pathogens among patients suffering from acute community-acquired diarrhea using multivariable logistic regression models in two Lebanese tertiary health care settings.**

| | Diarrheagenic Escherichia coli / Shigella spp. | | | | | | | | | | Other pathogenic bacteria | | | | | |
| --- | --- | --- | --- | --- | --- | --- | --- | --- | --- | --- | --- | --- | --- | --- | --- | --- |
| | EPEC | | EAEC | | ETEC | | EIEC/ Shigella spp. | | STEC | | Campylobacter spp. | | Salmonella spp. | | Clostridium difficile | |
| | Adjusted OR (IC95%) | P | Adjusted OR (IC95%) | P | Adjusted OR (IC95%) | P | Adjusted OR (IC95%) | P | Adjusted OR (IC95%) | P | Adjusted OR (IC95%) | P | Adjusted OR (IC95%) | P | Adjusted OR (IC95%) | P |
| **Age (continuous variable)** | | | 1.00 (0.98–1.01) | 0.87 | 1.01 (0.99–1.03) | 0.16 | | | | | | | | | | |
| **Sex** | | | | | | | | | | | | | | | | |
| Female[1] | | | | | | | | | | | | | | | | |
| Male | 0.72 (0.46–1.12) | 0.15 | | | **0.38 (0.20–0.70)** | **0.002** | | | | | | | | | | |
| **Region** | | | | | | | | | | | | | | | | |
| Urban[1] | | | | | | | | | | | | | | | | |
| Rural | | | 1.45 (0.76–2.80) | 0.26 | 1.31 (0.58–3.06) | 0.52 | | | | | 2.14 (0.84–6.64) | 0.14 | 0.50 (0.22–1.20) | 0.11 | | |
| **Using antiseptic products in washing** | | | | | | | | | | | | | | | | |
| Yes[1] | | | | | | | | | | | | | | | | |
| No | | | | | | | | | | | 2.42 (0.89–8.50) | 0.12 | | | | |
| **Swim (during the last 14 days)** | | | | | | | | | | | | | | | | |
| Yes | | | | | | | 1.57 (0.67–3.58) | 0.29 | | | | | | | | |
| No[1] | | | | | | | | | | | | | | | | |
| **Eating outside household** | | | | | | | | | | | | | | | | |
| 0-1/week[1] | | | | | | | | | | | | | | | | |
| >1/week | | | | | | | | | | | | | 1.67 (0.67–3.87) | 0.24 | | |
| **Drinking water** | | | | | | | | | | | | | | | | |
| Treated[1] | | | | | | | | | | | | | | | | |
| Untreated | 0.63 (0.39–1.01) | 0.05 | | | 0.55 (0.27–1.07) | 0.09 | | | | | | | | | | |
| **Season** | | | | | | | | | | | | | | | | |
| Summer[1] | | | | | | | | | | | | | | | | |
| Fall | | | | | **2.22 (1.12–4.49)** | **0.02** | 1.44 (0.73–2.88) | 0.30 | 0.39 (0.08–1.56) | 0.20 | 1.06 (0.43–2.67) | 0.90 | | | 0.27 (0.04–1.22) | 0.12 |
| Winter | | | | | **0.11 (0.03–0.37)** | **0.001** | **0.19 (0.06–0.53)** | **0.002** | $0.0\ (0.0–28{*}10^{49})$ | 0.99 | 1.02 (0.40–2.67) | 0.96 | | | 1.06 (0.32–3.52) | 0.92 |
| Spring | | | | | **0.23 (0.06–0.66)** | **0.01** | **0.20 (0.06–0.58)** | **0.006** | 0.65 (0.16–2.48) | 0.54 | 1.33 (0.52–3.45) | 0.56 | | | 1.59 (0.50–5.14) | 0.43 |
| **Abdominal pain** | | | | | | | | | | | | | | | | |
| Yes | | | 0.68 (0.42–1.09) | 0.11 | | | 0.65 (0.33–1.21) | 0.18 | **4.67 (1.44–17.9)** | **0.01** | 1.67 (0.87–3.17) | 0.12 | | | | |
| No[1] | | | | | | | | | | | | | | | | |
| **Vomiting** | | | | | | | | | | | | | | | | |
| Yes | | | 1.19 (0.72–1.96) | 0.49 | | | | | | | | | | | | |
| No[1] | | | | | | | | | | | | | | | | |

(Continued)

**Table 4.** (Continued)

| | Diarrheagenic Escherichia coli / Shigella spp. | | | | | | | | | | Other pathogenic bacteria | | | | | |
| --- | --- | --- | --- | --- | --- | --- | --- | --- | --- | --- | --- | --- | --- | --- | --- | --- |
| | EPEC | | EAEC | | ETEC | | EIEC/ Shigella spp. | | STEC | | Campylobacter spp. | | Salmonella spp. | | Clostridium difficile | |
| | Adjusted OR (IC95%) | P | Adjusted OR (IC95%) | P | Adjusted OR (IC95%) | P | Adjusted OR (IC95%) | P | Adjusted OR (IC95%) | P | Adjusted OR (IC95%) | P | Adjusted OR (IC95%) | P | Adjusted OR (IC95%) | P |
| **Fever** | | | | | | | | | | | | | | | | |
| Yes | | | 1.33 (0.81–2.19) | 0.27 | | | | | | | | | | | | |
| No[1] | | | | | | | | | | | | | | | | |

*Missing information has been omitted; Determinants of infection were predicted using multivariable logistic regression analysis.

medicines, widespread pollution, and a severe economic collapse and inflation [43–48]. This study is also unique in terms of the diagnostic technology, patient population (rural and urban), and assessment of patients with acute diarrhea of all ages in the community, while many of the previous studies only targeted hospitalized individuals. The findings revealed many possibilities to facilitate the control of diarrheal infections in the community. For example, our multivariable logistic regression analysis revealed that the occurrence of vomiting among diarrheal patients was significantly associated with the presence of enteric pathogens, particularly rotavirus A, which will raise the awareness and preparedness of clinicians to address these cases. Furthermore, it was notable that washing fruits and vegetables before consumption increased enteric infections, which is counterintuitive. However, this could be explained by the fact that most patients (45.9%, 156/340) used only water, which might be contaminated with enteric infectious agents. Lebanon is currently facing a water availability and pollution crisis due to the collapse of the economy. Indeed, recent hepatitis A and ongoing cholera outbreaks have been associated with contaminated water. Furthermore, most rivers and surface water resources were found to be contaminated with fecal microorganisms [38, 45, 46, 49–54], which facilitates the transmission of enteric pathogens in the Lebanese community via water (recreational or domestic) and the food production chain (irrigation water). This information adds urgency to address fecal pollution and improve the quality of water in Lebanon through provisions of access to safe water, sanitation, and sustained hygiene practices. Furthermore, there appears to be a need for outreach programs that target the education of the population on how to handle potentially contaminated waters and the associated ramifications.

This study has failed to identify the enteric pathogens associated in only 13.9% of the cases, which is lower than previous studies that used conventional tools [29–31] or that targeted specific enteric pathogens such as *Campylobacter* spp., *Cryptosporidium* spp., rotavirus, and norovirus [5, 32, 33, 35]. According to our findings, some of the most relevant enteric pathogens are currently neglected and underestimated by the Lebanese health authorities, including EAEC, ETEC, STEC, *Campylobacter* spp., *C. difficile*, norovirus, sapovirus, and *Cryptosporidium* spp. The prevalence of these enteric organisms ranges from 4% to 41.7% of the cases, but they are not routinely screened in clinical microbiology laboratories. This is resulting in many unidentified diarrheal cases and promoting mistreatment and the misuse of antimicrobials in Lebanon [29–31]. Therefore, our findings underlined the crucial need to widen test panels in clinical laboratories in Lebanon by adopting advanced diagnostic tools (e.g., multiplex PCR tests) or increasing the number of conventional tests (e.g., monoplex PCR assays, modified Ziehl–Neelsen staining). This will allow the identification of more enteric pathogens and

**Table 5. Determinants of viral and parasitic pathogens among patients suffering from acute community-acquired diarrhea using multivariable logistic regression models in two Lebanese tertiary health care settings.**

| | *Viruses* | | | | | | *Parasites* | | | | | |
|---|---|---|---|---|---|---|---|---|---|---|---|---|
| | **Rotavirus A** | | **Norovirus GI/GII** | | **Adenovirus** | | **Sapovirus** | | ***Cryptosporidium* spp.** | | ***Giardia duodenalis*** | |
| | **Adjusted OR (IC95%)** | *P* | **Adjusted OR (IC95%)** | *P* | **Adjusted OR (IC95%)** | *P* | **Adjusted OR (IC95%)** | *P* | **Adjusted OR (IC95%)** | *P* | **Adjusted OR (IC95%)** | *P* |
| **Age (continuous variable)** | **0.68 (0.50–0.88)** | **0.01** | 0.98 (0.93–1.01) | 0.23 | 0.80 (0.50–0.97) | 0.19 | | | 0.97 (0.91–1.01) | 0.27 | | |
| **Sex** | | | | | | | | | | | | |
| Female[1] | | | | | | | | | | | | |
| Male | | | | | | | | | | | 2.97 (0.87–13.6) | 0.11 |
| **Region** | | | | | | | | | | | | |
| Urban[1] | | | | | | | | | | | | |
| Rural | **8.51 (1.51–162)** | **0.04** | | | | | | | | | | |
| **Using antiseptic products in washing** | | | | | | | | | | | | |
| Yes[1] | | | | | | | | | | | | |
| No | 0.84 (0.32–2.32) | 0.73 | | | | | | | 5.46 (1.06–100) | 0.10 | | |
| **Swim (during the last 14 days)** | | | | | | | | | | | | |
| Yes | | | **3.33 (1.12–8.85)** | **0.02** | | | 3.18 (0.76–12.8) | 0.10 | | | | |
| No[1] | | | | | | | | | | | | |
| **Eating outside household** | | | | | | | | | | | | |
| 0-1/week[1] | | | | | | | | | | | | |
| >1/week | 0.55 (0.19–1.40) | 0.23 | | | | | 2.09 (0.63–6.43) | 0.21 | | | 2.75 (0.79–8.82) | 0.09 |
| **Drinking water** | | | | | | | | | | | | |
| Treated[1] | | | | | | | | | | | | |
| Untreated | | | | | | | | | | | | |
| **Season** | | | | | | | | | | | | |
| Summer[1] | | | | | | | | | | | | |
| Fall | 1.17 (0.47–2.94) | 0.74 | | | 4.29 (1.08–28.6) | 0.07 | 2.22 (0.63–8.95) | 0.23 | 2.91 (0.95–10.9) | 0.08 | | |
| Winter | **5.15 (2.21–12.7)** | **<0.001** | | | 2.24 (0.46–16.1) | 0.34 | 0.70 (0.09–4.30) | 0.70 | 0.64 (0.12–3.05) | 0.57 | | |
| Spring | 1.22 (0.50–3.05) | 0.66 | | | 0.53 (0.02–5.64) | 0.61 | $0.0 (0.0–92*10^{23})$ | 0.99 | 0.74 (0.14–3.02) | 0.70 | | |
| **Abdominal pain** | | | | | | | | | | | | |
| Yes | | | | | | | 0.28 (0.04–1.05) | 0.10 | | | | |
| No[1] | | | | | | | | | | | | |
| **Vomiting** | | | | | | | | | | | | |
| Yes | **3.12 (1.55–6.56)** | **0.002** | | | | | | | | | | |
| No[1] | | | | | | | | | | | | |
| **Fever** | | | | | | | | | | | | |

(*Continued*)

**Table 5.** (Continued)

| | Viruses | | | | | | Parasites | | | | | | |
|---|---|---|---|---|---|---|---|---|---|---|---|---|---|
| | Rotavirus A | | Norovirus GI/GII | | Adenovirus | | Sapovirus | | Cryptosporidium spp. | | Giardia duodenalis | |
| | Adjusted OR (IC95%) | P | Adjusted OR (IC95%) | P | Adjusted OR (IC95%) | P | Adjusted OR (IC95%) | P | Adjusted OR (IC95%) | P | Adjusted OR (IC95%) | P |
| Yes | 1.09 (0.56–2.14) | 0.80 | **0.41 (0.17–0.96)** | **0.04** | | | | | | | **0.25 (0.07–0.79)** | **0.02** |
| No[1] | | | | | | | | | | | | |

*Missing information has been omitted; Determinants of infection were predicted using multivariable logistic regression analysis.

enhance the understanding of the etiology and epidemiology of diarrhea, which will, in turn, improve patient outcomes and disease control. Although the FilmArray Gastrointestinal Panel assay is an easy-to-use and rapid test with high accuracy that allows the detection of the most common enteric pathogens [55, 56], it should be highlighted that even this panel does not identify all known infectious agents of diarrheal diseases. We argue the need for further tests to identify enteric pathogens, especially in the cases where the panel failed to detect an etiologic agent. The latter might include *Listeria monocytogenes*, *Aeromonas* spp., *Blastocystis* spp., *Cystoisospora belli*, roundworms, and tapeworms. Despite the aforementioned limitations in the panel, this study demonstrated that EAEC infections were the most common among diarrheic patients in the community in North Lebanon, independent of their age, sex, and region. Even though the pathogenesis of EAEC is not well understood, this bacterium represents an emerging pathogen that is increasingly recognized as a cause of acute diarrhea among children and adults [57, 58]. Diarrheal patients carrying EAEC showed a higher percentage of vomiting and fever compared to EAEC-negative individuals; however, this observation was not statistically significant.

The second leading cause of diarrhea in our study (i.e., EPEC) is also rarely tested in clinical laboratories in Lebanon. When tested, slide agglutination assays are performed on *E. coli* isolates in children younger than two years old and are associated with limited sensitivity and specificity [59]. For many years, EPEC has been associated with infantile watery diarrhea in developing countries [60]. In our previous retrospective study on bacterial enteric pathogens in children carried out between 2009 and 2015 at the Nini hospital, we described that EPEC was more prevalent than *Salmonella* spp. And *Shigella* spp. And showed a higher percentage of resistance to third-generation cephalosporins [61]. Again, the latter corroborated the need to adopt an approach that targets multiple pathogens when investigating diarrheal diseases in LMICs.

Multivariable logistic regression analysis showed an association between diarrheagenic *E. coli* (i.e., an increase in ETEC infections among EAEC, and EPEC infections and vice versa). Moreover, there was a significant association between ETEC and EIEC/*Shigella* spp. Infections. ETEC was the third most common diarrheagenic *E. coli*, followed by EIEC/*Shigella* spp. And STEC. Notably, there were no seasonal variations associated with EAEC, EPEC, and STEC infections (P > 0.05). However, ETEC and EIEC/*Shigella* spp. Infections were significantly more likely to occur in the summer season in comparison to winter and spring. The bivariate analysis showed that patients who swam during the last 14 days before testing had 2-fold more EIEC/*Shigella* spp. Infections than non-swimmers, which partially explains the summer trend, especially when considering the aforementioned surface water pollution in Lebanon [45, 48, 62]. Interestingly, diarrheagenic *E. coli* are also strongly associated with viral

**Table 6.** Association between enteric pathogens among patients suffering from acute community-acquired diarrhea using multivariable logistic regression models in two Lebanese tertiary health care settings.

| | Model 1[i] | | | | Model 2[ii] | | | |
|---|---|---|---|---|---|---|---|---|
| | adj. OR | 95% CI | P-value | adj. P-value[§] | adj. OR | 95% CI | P-value | adj. P-value[§] |
| *EPEC* | | | | | | | | |
| **EAEC** | **2.05** | **1.29–3.27** | **0.002** | **0.021** | **2.05** | **1.31–3.22** | **0.002** | **0.018** |
| **ETEC** *lt/st* | **2.08** | **1.18–3.70** | **0.011** | **0.043** | **2.22** | **1.28–3.90** | **0.005** | **0.027** |
| **EIEC/***Shigella* **spp.** | 1.50 | 0.83–2.69 | 0.177 | 0.324 | | | | |
| *Campylobacter* **spp.** | 1.00 | 0.53–1.87 | 0.999 | 1.00 | | | | |
| *Salmonella* **spp.** | 1.23 | 0.56–2.64 | 0.592 | 0.781 | | | | |
| *Clostridium difficile* **toxin A/B** | 1.44 | 0.55–3.68 | 0.448 | 0.626 | | | | |
| **Rotavirus A** | 1.79 | 1.07–2.99 | 0.025 | 0.084 | 1.68 | 1.02–2.75 | 0.040 | 0.113 |
| **Norovirus GI/GII** | 0.85 | 0.35–1.97 | 0.705 | 0.862 | | | | |
| *Cryptosporidium* **spp.** | 1.65 | 0.68–3.98 | 0.264 | 0.403 | | | | |
| *EAEC* | | | | | | | | |
| **EPEC** | **2.03** | **1.28–3.24** | **0.003** | **0.021** | **2.03** | **1.28–3.21** | **0.003** | **0.021** |
| **ETEC** *lt/st* | **2.80** | **1.56–5.09** | **0.001** | **0.016** | **2.73** | **1.53–4.95** | **0.001** | **0.016** |
| **EIEC/***Shigella* **spp.** | 1.73 | 0.94–3.18 | 0.077 | 0.165 | 1.58 | 0.87–2.87 | 0.132 | 0.263 |
| *Campylobacter* **spp.** | 1.85 | 0.98–3.54 | 0.059 | 0.151 | 1.82 | 0.97–3.44 | 0.064 | 0.155 |
| *Salmonella* **spp.** | 1.54 | 0.70–3.33 | 0.271 | 0.410 | | | | |
| *Clostridium difficile* **toxin A/B** | 1.20 | 0.45–3.13 | 0.713 | 0.862 | | | | |
| **Rotavirus A** | **2.44** | **1.45–4.15** | **0.001** | **0.016** | **2.33** | **1.39–3.91** | **0.001** | **0.018** |
| **Norovirus GI/GII** | **3.50** | **1.47–8.85** | **0.006** | **0.028** | **3.55** | **1.50–8.94** | **0.005** | **0.027** |
| *Cryptosporidium* **spp.** | 0.57 | 0.21–1.43 | 0.247 | 0.383 | | | | |
| *ETEC lt/st* | | | | | | | | |
| **EPEC** | **2.09** | **1.19–3.73** | **0.011** | **0.043** | **2.07** | **1.18–3.67** | **0.012** | **0.043** |
| **EAEC** | **2.88** | **1.60–5.25** | **<0.001** | **0.016** | **2.70** | **1.53–4.83** | **0.001** | **0.016** |
| **EIEC/***Shigella* **spp.** | **2.35** | **1.23–4.44** | **0.009** | **0.035** | **2.49** | **1.33–4.60** | **0.004** | **0.026** |
| *Campylobacter* **spp.** | 1.00 | 0.44–2.12 | 0.995 | 1.00 | | | | |
| *Salmonella* **spp.** | 0.61 | 0.19–1.63 | 0.358 | 0.511 | | | | |
| *Clostridium difficile* **toxin A/B** | 0.72 | 0.16–2.39 | 0.626 | 0.811 | | | | |
| **Rotavirus A** | **0.29** | **0.13–0.60** | **0.002** | **0.018** | **0.31** | **0.14–0.64** | **0.003** | **0.021** |
| **Norovirus GI/GII** | 0.65 | 0.21–1.75 | 0.422 | 0.595 | | | | |
| *Cryptosporidium* **spp.** | 1.19 | 0.38–3.26 | 0.748 | 0.888 | | | | |
| *EIEC/Shigella* **spp.** | | | | | | | | |
| **EPEC** | 1.49 | 0.83–2.67 | 0.180 | 0.324 | | | | |
| **EAEC** | 1.72 | 0.94–3.16 | 0.081 | 0.165 | 1.93 | 1.08–3.49 | 0.027 | 0.087 |
| **ETEC** *lt/st* | **2.32** | **1.21–4.38** | **0.010** | **0.041** | **2.46** | **1.31–4.59** | **0.005** | **0.027** |
| *Campylobacter* **spp.** | 0.96 | 0.42–2.06 | 0.928 | 0.994 | | | | |
| *Salmonella* **spp.** | **0.20** | **0.03–0.73** | **0.036** | 0.104 | 0.20 | 0.03–0.71 | 0.034 | 0.099 |
| *Clostridium difficile* **toxin A/B** | 0.37 | 0.06–1.44 | 0.211 | 0.354 | | | | |
| **Rotavirus A** | **0.32** | **0.14–0.68** | **0.005** | **0.027** | **0.32** | **0.14–0.68** | **0.005** | **0.027** |
| **Norovirus GI/GII** | 1.88 | 0.72–4.69 | 0.183 | 0.324 | | | | |
| *Cryptosporidium* **spp.** | 2.59 | 0.94–6.76 | 0.056 | 0.146 | 2.43 | 0.90–6.14 | 0.067 | 0.155 |
| *Campylobacter* **spp.** | | | | | | | | |
| **EPEC** | 1.01 | 0.53–1.89 | 0.986 | 1.00 | | | | |
| **EAEC** | 1.83 | 0.96–3.50 | 0.066 | 0.155 | 1.73 | 0.96–3.17 | 0.070 | 0.160 |

*(Continued)*

**Table 6.** (Continued)

| | Model 1[i] | | | | Model 2[ii] | | | |
|---|---|---|---|---|---|---|---|---|
| | adj. OR | 95% CI | P-value | adj. P-value[§] | adj. OR | 95% CI | P-value | adj. P-value[§] |
| **ETEC** *lt/st* | 0.93 | 0.41–1.99 | 0.852 | 0.942 | | | | |
| **EIEC/*Shigella* spp.** | 0.88 | 0.38–1.90 | 0.759 | 0.889 | | | | |
| *Salmonella* **spp.** | 0.16 | 0.01–0.79 | 0.076 | 0.165 | 0.17 | 0.01–0.81 | 0.081 | 0.165 |
| *Clostridium difficile* **toxin A/B** | 0.29 | 0.02–1.49 | 0.239 | 0.379 | | | | |
| **Rotavirus A** | 0.96 | 0.47–1.90 | 0.910 | 0.987 | | | | |
| **Norovirus GI/GII** | 0.81 | 0.22–2.30 | 0.711 | 0.862 | | | | |
| *Cryptosporidium* **spp.** | 0.27 | 0.01–1.38 | 0.212 | 0.354 | 0.25 | 0.01–1.26 | 0.186 | 0.324 |
| *Salmonella* **spp.** | | | | | | | | |
| **EPEC** | 1.28 | 0.58–2.77 | 0.541 | 0.731 | | | | |
| **EAEC** | 1.65 | 0.73–3.69 | 0.227 | 0.369 | | | | |
| **ETEC** *lt/st* | 0.61 | 0.19–1.64 | 0.356 | 0.511 | | | | |
| **EIEC/*Shigella* spp.** | **0.19** | **0.03–0.68** | **0.029** | 0.090 | 0.20 | 0.03–0.70 | 0.032 | 0.097 |
| *Campylobacter* **spp.** | 0.13 | 0.01–0.67 | 0.054 | 0.146 | 0.16 | 0.01–0.78 | 0.076 | 0.165 |
| *Clostridium difficile* **toxin A/B** | 0.77 | 0.11–2.99 | 0.736 | 0.881 | | | | |
| **Rotavirus A** | **0.22** | **0.06–0.60** | **0.007** | **0.034** | 0.28 | 0.08–0.74 | 0.021 | 0.073 |
| **Norovirus GI/GII** | 0.24 | 0.01–1.32 | 0.186 | 0.324 | | | | |
| *Cryptosporidium* **spp.** | 1.17 | 0.25–3.93 | 0.816 | 0.934 | | | | |
| *Clostridium difficile* **toxin A/B** | | | | | | | | |
| **EPEC** | 1.37 | 0.53–3.51 | 0.506 | 0.691 | | | | |
| **EAEC** | 1.24 | 0.45–3.32 | 0.667 | 0.847 | | | | |
| **ETEC** *lt/st* | 0.70 | 0.15–2.36 | 0.596 | 0.781 | | | | |
| **EIEC/*Shigella* spp.** | 0.39 | 0.06–1.52 | 0.236 | 0.379 | | | | |
| *Campylobacter* **spp.** | 0.29 | 0.02–1.51 | 0.242 | 0.380 | 0.28 | 0.02–1.40 | 0.222 | 0.366 |
| *Salmonella* **spp.** | 0.79 | 0.12–3.04 | 0.763 | 0.889 | | | | |
| **Rotavirus A** | 0.90 | 0.29–2.47 | 0.847 | 0.942 | | | | |
| **Norovirus GI/GII** | 1.41 | 0.20–5.89 | 0.673 | 0.847 | | | | |
| *Cryptosporidium* **spp.** | 2.49 | 0.53–8.65 | 0.185 | 0.324 | | | | |
| **Rotavirus A** | | | | | | | | |
| **EPEC** | **1.80** | **1.08–3.01** | **0.025** | 0.084 | 1.79 | 1.07–2.99 | 0.026 | 0.084 |
| **EAEC** | **2.51** | **1.48–4.31** | **0.001** | **0.016** | **2.49** | **1.48–4.25** | **0.001** | **0.016** |
| **ETEC** *lt/st* | **0.29** | **0.13–0.60** | **0.001** | **0.018** | **0.29** | **0.13–0.60** | **0.002** | **0.018** |
| **EIEC/*Shigella* spp.** | **0.31** | **0.13–0.66** | **0.004** | **0.026** | **0.31** | **0.13–0.66** | **0.004** | **0.026** |
| *Campylobacter* **spp.** | 0.93 | 0.45–1.84 | 0.837 | 0.942 | | | | |
| *Salmonella* **spp.** | **0.22** | **0.06–0.61** | **0.008** | **0.034** | **0.22** | **0.06–0.61** | **0.008** | **0.034** |
| *Clostridium difficile* **toxin A/B** | 0.87 | 0.28–2.39 | 0.794 | 0.917 | | | | |
| **Norovirus GI/GII** | 0.32 | 0.09–0.94 | 0.056 | 0.146 | 0.32 | 0.09–0.94 | 0.055 | 0.146 |
| *Cryptosporidium* **spp.** | 0.96 | 0.32–2.54 | 0.931 | 0.994 | | | | |
| **Norovirus GI/GII** | | | | | | | | |
| **EPEC** | 0.84 | 0.35–1.97 | 0.697 | 0.862 | | | | |
| **EAEC** | **3.44** | **1.44–8.74** | **0.007** | **0.032** | **3.31** | **1.46–8.01** | **0.005** | **0.027** |
| **ETEC** *lt/st* | 0.68 | 0.22–1.81 | 0.461 | 0.636 | | | | |
| **EIEC/*Shigella* spp.** | 1.95 | 0.74–4.87 | 0.162 | 0.316 | | | | |
| *Campylobacter* **spp.** | 0.90 | 0.25–2.57 | 0.853 | 0.942 | | | | |
| *Salmonella* **spp.** | 0.35 | 0.02–1.85 | 0.320 | 0.468 | | | | |

*(Continued)*

**Table 6.** (Continued)

| | Model 1[i] | | | | Model 2[ii] | | | |
|---|---|---|---|---|---|---|---|---|
| | adj. OR | 95% CI | P-value | adj. P-value[§] | adj. OR | 95% CI | P-value | adj. P-value[§] |
| *Clostridium difficile* toxin A/B | 1.63 | 0.23–6.95 | 0.555 | 0.742 | | | | |
| **Rotavirus A** | 0.36 | 0.10–1.04 | 0.082 | 0.165 | 0.36 | 0.10–0.97 | 0.067 | 0.155 |
| *Cryptosporidium* spp. | 0.00 | $0-2*10^{22}$ | 0.990 | 1.00 | 0.00 | $0-4*10^{22}$ | 0.990 | 1.00 |
| *Cryptosporidium* spp. | | | | | | | | |
| **EPEC** | 1.61 | 0.67–3.92 | 0.285 | 0.421 | | | | |
| **EAEC** | 0.60 | 0.23–1.49 | 0.284 | 0.421 | | | | |
| **ETEC** *lt/st* | 1.10 | 0.35–3.07 | 0.862 | 0.944 | | | | |
| **EIEC/***Shigella* **spp.** | 2.44 | 0.87–6.46 | 0.079 | 0.165 | 2.32 | 0.90–5.55 | 0.066 | 0.155 |
| *Campylobacter* **spp.** | 0.26 | 0.01–1.33 | 0.200 | 0.344 | 0.24 | 0.01–1.18 | 0.166 | 0.319 |
| *Salmonella* **spp.** | 1.35 | 0.30–4.47 | 0.658 | 0.844 | | | | |
| *Clostridium difficile* **toxin A/B** | 2.48 | 0.53–8.56 | 0.186 | 0.324 | | | | |
| **Rotavirus A** | 0.96 | 0.32–2.61 | 0.942 | 0.997 | | | | |
| **Norovirus GI/GII** | 0.00 | $0-3*10^{10}$ | 0.990 | 1.00 | 0.00 | $0-5*10^{22}$ | 0.990 | 1.00 |

[i]In Model 1, selected enteric pathogens (Prevalence>5%) were entered in the model as explanatory variables.

[ii]In Model 2, a backward logistic regression model was created.

[§]P-values were adjusted according to (Benjamini and Hochberg, 1995).

infections. Our findings confirmed previous observations that EAEC and rotavirus mixed infections were frequent among children [63] and patients having severe diarrhea [64]. However, to our knowledge, we reported for the first time a tendency for EAEC and norovirus co-infections. While this observation might be specific to the tested Lebanese population, it still highlights the importance of screening for multiple diarrheal agents.

Although the Lebanese health authorities introduced rotavirus vaccines to the private market in 2006, the prevalence of rotavirus infections is still high in the Lebanese community, particularly among children younger than 5 years [65]. In our study, rotavirus A represented the third leading cause of diarrhea. Our results corroborated previous data that showed that group A rotavirus was common (~30%) in the Lebanese community [29, 30, 35, 66], affecting around 28% and 75% of children with acute gastroenteritis who are younger than five and two years old, respectively [67]. We found that rotavirus A infections were more likely to occur among patients living in rural areas compared to those from urban areas and were significantly associated with vomiting. Moreover, we found that rotavirus-positive cases were more frequent during the winter in comparison to the summer season. These findings could be explained by the fact that more flood events occur in the Akkar governorate (i.e., the rural area) in comparison to the urban area during the winter season. This is important, given that well water is the most common source of drinking water for most households in the villages of Akkar, followed by trucked water. Flooding from polluted rivers can contaminate these water sources. Furthermore, this overcrowded rural region has the highest poverty rate in Lebanon and suffers from the lack of appropriate sewer systems and poor water, sanitation, and hygiene services. Taken together, we hypothesize that the increase in the prevalence of rotavirus infections during the winter season is possibly related to the contamination of water sources by the virus as well as the close contact between children at home and in childcare facilities during the cold weather. Despite the absence of any information about the vaccination status of the participants, it is worth mentioning that the community of Akkar has a low socioeconomic status, weak health

infrastructure, poor education and awareness of infectious diseases, and less adherence to the national immunization calendar. The latter is notable when considering that rotavirus vaccines are voluntary and not covered by the Lebanese National Immunization Program of the Ministry of Public Health [68]. Indeed, fewer than 30% of children in Lebanon receive the rotavirus vaccine [65–67], and this percentage is likely lower in the Akkar governorate. Since previous data showed that the larger prevalence of rotavirus infection was found among non-vaccinated patients in the Lebanese community [29, 66, 69], our data highlighted the critical need to include rotavirus vaccines in the Lebanese National Immunization Program and implement mass vaccination campaigns to decrease the burden of rotavirus in the community. It would be amiss not to mention that Lebanon is currently hosting approximately 2 million Syrian and Palestinian refugees, a large population of whom suffer from poverty and low- or no vaccination and might drive the transmission of disease in the community [36–38].

Around 10% of cases in our study were attributed to *Salmonella* infections. This was not surprising, because *Salmonella* is endemic in Lebanon and can induce gastroenteritis as well as invasive infections such as bacteremia and systemic disease among high-risk individuals [70]. In comparison, markedly limited data exist on the contribution of norovirus, *Campylobacter* spp., and *C. difficile* to the burden of diarrheal disease in Lebanon. The only two available reports on norovirus showed a prevalence ranging between 6% and 11% among children with acute gastroenteritis [35, 71], while neither the prevalence nor the burden of campylobacteriosis are well characterized in humans and animals [72]. However, the few available reports have revealed a high prevalence of *Campylobacter* infections associated with multidrug-resistant isolates [32–34]. For example, we reported the emergence of a multidrug-resistant, $bla_{OXA-61}$-producing, and hypervirulent *Campylobacter coli* isolated from the stool of a Lebanese newborn with severe diarrhea [73]. Similarly, although there is a paucity of data on *C. difficile* in Lebanon, our results corroborated the only existing study in Lebanon, which showed a non-negligible percentage of *C. difficile* infections (5.5%) among patients presenting to the emergency department of a tertiary care center [74]. Due to the low number of cases, we were unable to identify the risk factors and effective treatment strategies of this emerging pathogen. Our current data, combined with previous reports, indicate the need to incorporate routine screening tests for these bacteria to allow early diagnosis and to monitor the pathogens and their burden.

We identified two cases of cholera in our study, one patient in Ain Al Zeit (a town in the Akkar governorate situated near the Lebanese Syrian border) and the other in Kobbeh (an overcrowded suburb in Tripoli, the North governorate). Over the last few years, rare sporadic incidences of Non-O1, non-O139 *V. cholerae* have been reported in Lebanon after the 2015 Lebanese trash crisis [75]. Currently, the Lebanese public health authorities are actively inspecting cholera cases in the Lebanese governorates that are thought to have spilled over from neighboring Syria, carried by the transboundary movement of Syrian refugees hosted in Lebanon [18]. As of October 19, there have been 169 cholera cases, particularly in the Akkar and Baalbek Hermel governorates (https://www.thenationalnews.com/mena/lebanon/2022/10/19/lebanons-cholera-outbreak-kills-five-as-case-numbers-surge/), and five deaths. Taken together, our data support that cholera might have been present in the Lebanese territories, and its burden was probably underestimated due the lack of appropriate diagnostic tools, even though official records have not declared any cholera cases since 1993. However, given the status of water quality and sanitation systems in both Lebanon and Syria, we believe that a cholera outbreak was imminent and would affect both countries. Regardless, early detection might have mobilized governmental resources to avoid an outbreak, whether by closely monitoring water quality in the most disenfranchised areas and/or by providing preemptive vaccination.

The most frequent intestinal parasites detected in this study were *Cryptosporidium* spp. And *G. duodenalis*. These two protozoans were previously reported in higher percentages among children from the same geographic area (Akkar and North governates) [4]. In this study, a relatively similar prevalence of cryptosporidiosis was detected compared to earlier investigations. Although *Cryptosporidium* is among the most common intestinal protozoan pathogens in the Middle East region [76], with a prevalence reaching up to 10% in epidemiologic studies conducted in Lebanon [4, 5, 28, 77], it is still not a priority to either the Lebanese Ministry of Public Health or medical laboratories. The highest prevalence of cryptosporidiosis in this study has been observed during the fall season, which may be potentially associated with the increase in children returning to childcare facilities and schools and/ or the increase in flood events which occur in early September.

In comparison to our study, a much higher prevalence of *G. duodenalis* (29%) was previously reported among symptomatic and asymptomatic children [4]. This difference in prevalence could be related to many factors, such as the population age and the exclusion of asymptomatic or hospitalized patients from our patient population. In our study, neither other PCR assays nor direct light microscopy were performed, which prevented the detection of other endemic intestinal parasites such as *Blastocystis* spp. and *Dientamoeba fragilis*. Interestingly, none of the samples were positive for *E. histolytica* and *C. cayetanensis*. Taken together, our data highlight yet again the need to more actively monitor parasites in Lebanon, especially among children younger than 5 years old with diarrhea [4, 5].

Despite the inherent limitations of our approach that included not screening for certain pathogens and the nature of the study population (mainly children from rural areas), we strongly believe that this study is highly valuable, identifying multiple etiologic agents and overlooked pathogens, and the susceptibility of disenfranchised populations to enteric infections. These populations can also be indicators/reservoirs of infections in LMICs. For example, although the Epidemiological Surveillance Program of the Lebanese Ministry of Public Health has not reported any *V. cholerae* over the last three decades, we described the emergence of cholera cases in both Akkar and North governorates. These findings are extremely critical for preemptive disease control, particularly when considering the unfolding of the cholera outbreak and other water-borne and foodborne enteric in Lebanon [78].

## 5. Conclusions

This is the first study performed in Lebanon that reports the prevalence, trends, and clinical data associated with enteric pathogens in the community. A high prevalence of bacterial, viral, and protozoan agents non-routinely tested in the Lebanese clinical laboratories was found. Also, to our knowledge, this is the first investigation reporting *P. shigelloides*, sapovirus, and astrovirus infections in Lebanon. Although the use of molecular tools for the identification of diarrheal etiologies is relatively expensive, this approach confers an increased sensitivity and specificity to detect the most enteric pathogens and help healthcare professionals to mobilize earlier and prescribe more appropriate interventions to their patients, reduce the overuse of antimicrobials, and prevent outbreaks. Our findings also highlight the urgent need for close collaborations between the Lebanese government and international stakeholders and funding agencies to implement new relevant policies, legislations, and interventions to prevent the spread of infectious diseases in Lebanon and beyond.

## Supporting information

**S1 Fig. Pathogen association patterns among patients suffering from acute community-acquired diarrhea in two Lebanese tertiary healthcare settings.** * ADV: Adenovirus F40/41;

AST: Astrovirus; CAMP: *Campylobacter* spp.; CDIF: *Clostridium difficile*; CRYP: *Cryptosporidium* spp.; EAEC: Enteroaggregative *Escherichia coli*; EIEC: Enteroinvasive *E. coli*/*Shigella* spp.; EPEC: Enteropathogenic *E. coli*; ETEC: Enterotoxigenic *E. coli*; GIA: *Giardia duodenalis*; NRV: Norovirus GI/GII; PS: *Plesiomonas shigelloides*; RTV: Rotavirus A; SAL: *Salmonella* spp.; SPV: Sapovirus (I, II, IV, and V); STEC: Shiga toxin-producing *E. coli*; VIB: *Vibrio cholerae*. (TIF)

## Acknowledgments

We thank Taha Abdou and Majd Mouzawak for their helpful technical assistance.

## Author Contributions

**Conceptualization:** Marwan Osman, Monzer Hamze.

**Data curation:** Marwan Osman, Issmat I. Kassem, Fouad Dabboussi, Kevin J. Cummings, Monzer Hamze.

**Formal analysis:** Marwan Osman, Issmat I. Kassem, Fouad Dabboussi, Kevin J. Cummings, Monzer Hamze.

**Funding acquisition:** Monzer Hamze.

**Investigation:** Marwan Osman, Issmat I. Kassem, Fouad Dabboussi, Kevin J. Cummings, Monzer Hamze.

**Methodology:** Marwan Osman, Fouad Dabboussi, Monzer Hamze.

**Project administration:** Marwan Osman, Monzer Hamze.

**Resources:** Monzer Hamze.

**Software:** Marwan Osman.

**Supervision:** Kevin J. Cummings, Monzer Hamze.

**Validation:** Marwan Osman, Monzer Hamze.

**Visualization:** Marwan Osman, Issmat I. Kassem, Kevin J. Cummings.

**Writing – original draft:** Marwan Osman.

**Writing – review & editing:** Issmat I. Kassem, Fouad Dabboussi, Kevin J. Cummings, Monzer Hamze.

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
