## [Decision Letter · Decision Letter 0]

2 Feb 2023

PONE-D-22-30455The prevalence, clinical characterization, and seasonal trends of enteric pathogens among patients with acute community-acquired diarrhea in North LebanonPLOS ONE

Dear Dr. Hamze,

Thank you for submitting your manuscript to PLOS ONE. After careful consideration, we feel that it has merit but does not fully meet PLOS ONE’s publication criteria as it currently stands. Therefore, we invite you to submit a revised version of the manuscript that addresses the points raised during the review process.

Please, consider a more appropriate title (not so local) as it could decrease interest in your reading (in fact it was very difficult to find reviewers, who surely only read the title of the work). It is an interesting work that with a broader title would be attractive. Additionally, the labels of the figures should be larger and in bold so that they stand out.

We look forward to receiving your revised manuscript.

Kind regards,

Fernando Navarro-Garcia

Academic Editor

PLOS ONE

Journal Requirements:

Availability statement, you have not specified where the minimal data set underlying the results described in your manuscript can be found. PLOS defines a study's minimal data set as the underlying data used to reach the conclusions drawn in the manuscript and any additional data required to replicate the reported study findings in their entirety. All PLOS journals require that the minimal data set be made fully available. For more information about our data policy, please see http://journals.plos.org/plosone/s/data-availability.

Reviewers' comments:

Reviewer's Responses to Questions

**Comments to the Author**

1. Is the manuscript technically sound, and do the data support the conclusions?

Reviewer #1: Yes

Reviewer #2: Yes

2. Has the statistical analysis been performed appropriately and rigorously? 

Reviewer #1: Yes

Reviewer #2: Yes

3. Have the authors made all data underlying the findings in their manuscript fully available?

Reviewer #1: Yes

Reviewer #2: Yes

4. Is the manuscript presented in an intelligible fashion and written in standard English?

Reviewer #1: Yes

Reviewer #2: Yes

5. Review Comments to the Author

Reviewer #1: The manuscript of Osman et al describes a very well designed epidemiological study of the etiology of community acquired acute diarrhea in North Lebanon. The work enrolled 360 patients of different ages and included evaluations of clinical outcomes and risk factors. The data obtained is of high interest not only for the Lebanese health system but also for the researchers in the field. The authors found a high frequency of diarrheagenic Escherichia coli pathotypes, specifically EAEC and EPEC, and rotavirus A. The data is adequately presented and analysed, including multivariable logistic regression analyses. A large number of mixed infections was detected and the composition of them is presented. This could be highlighted in the discussion section in the view of the severity or associated clinical symptoms of these case. Also, two other limitations of the study should be considered in the discussion: the wide range of the patients ages (1-91 years old) and the lack of EPEC and EAEC sub-typing (typical and atypical isolates).

Minor points:

- page 4, 2nd paragraph: Important studies from the group of Richard Guerrant conducted in Fortaleza, Brazil, should be cited to support the information regarding the impaired cognitive consequences of diarrhea in children leaving in low income areas.

- page 5, lines 19-25: The text fragment starting as "For example, 11% of hospitalized...", finishing as "acute gastroenteritis, respectively [31]", would be more appropriate in the Discussion section.

- page 6: What does WASH stand for?

- Figure 2: Rearrange the bars in descending order of prevalence.

- Figure 3B: Not all types of co-infections detected are represented. This part of Fig. 3 could be presented as Supplementary Material describing the composition (enteric pathogens) found in all mixed infections. This is a relevant result to be presented.

- Figure 3B: Detection of co-infections is a common finding in epidemiological studies of the etiology of acute diarrhea in low income countries. However, co-infections of 4 enteric pathogens such as the one presented here (EPEC, EAEC, ETEC and EIEC) is notable, if real. Did the authors investigated the accuracy of such infections in 6 patients? Fecal culture and further colony identification? Separate PCR detection?

- page 14, line 12: Please change the reference 55 for other relevant references to support the point discussed regarding the pathogenesis of EAEC. Several reviews on EAEC pathogenesis would fit better here.

- page 14, line 16: The authors could discusses here the lack of sensitivity when using serum agglutination detection the so-called classical EPEC O serogroups.

Reviewer #2: Osman et al.

In this work that authors used robust molecular diagnostic methods that target multiple etiologic agents increases the sensitivity and specificity of detection for most enteric pathogens. To better understand the epidemiology of enteric infections, they aimed to determine the prevalence of enteric pathogens, identify risk factors and seasonal variations, and describe associations between pathogens among diarrheic patients in a Lebanese community. The authors found a high prevalence of enteric pathogens, with a predominance of mixed infections. Their statistical models revealed a strong relationship between different diarrheagenic Escherichia coli. Rotavirus A was significantly more likely to occur in the winter season in comparison to the summer. Furthermore, they found that living in rural areas and swimming are considered significant risk factors associated with rotavirus A and norovirus GI/GII infections, respectively. It is a very interesting and well-designed study and the authors’ conclusions are supported by their evidence. I have only minor concerns:

1) Discussion section, third paragraph: EPEC does not cause dysentery.

2) In fact, the reference 56 does not mention the word “dysentery”.

3) Discussion section, third paragraph, penultimate sentence: rewrite that sentence so as not to give the impression that the authors' work includes bacterial resistance experiments (clearly writing that this is a correlation with reference 57).

4) The conclusion section is too long and redundant, reduce it and try merging it with the conclusion as the last paragraph (or reduce the conclusion as well, it is redundant).

6. PLOS authors have the option to publish the peer review history of their article (what does this mean?). If published, this will include your full peer review and any attached files.

Reviewer #1: No

Reviewer #2: No

---

## [Author Response · Author response to Decision Letter 0]

5 Feb 2023

Editor and Reviewers’ comments:

Editor:

Comment 1. Consider a more appropriate title (not so local) as it could decrease interest in your reading (in fact it was very difficult to find reviewers, who surely only read the title of the work). It is an interesting work that with a broader title would be attractive.

REPLY: Done.

The new title is “The indelible toll of enteric pathogens: Prevalence, clinical characterization, and seasonal trends in patients with acute community-acquired diarrhea in disenfranchised communities”.

Comment 2. The labels of the figures should be larger and in bold so that they stand out.

REPLY: Done. We modified the figures according to your suggestions. However, it is worth mentioning that we are unable to increase the labels more than the actual size and maintain a high resolution because we will obtain an error.

Comment 3. Please ensure that your manuscript meets PLOS ONE's style requirements, including those for file naming. The PLOS ONE style templates can be found at

REPLY: Done.

Comment 4. Availability statement, you have not specified where the minimal data set underlying the results described in your manuscript can be found. PLOS defines a study's minimal data set as the underlying data used to reach the conclusions drawn in the manuscript and any additional data required to replicate the reported study findings in their entirety. All PLOS journals require that the minimal data set be made fully available. For more information about our data policy, please see http://journals.plos.org/plosone/s/data-availability. Upon re-submitting your revised manuscript, please upload your study’s minimal underlying data set as either Supporting Information files or to a stable, public repository and include the relevant URLs, DOIs, or accession numbers within your revised cover letter. For a list of acceptable repositories, please see http://journals.plos.org/plosone/s/data-availability#loc-recommended-repositories. Any potentially identifying patient information must be fully anonymized. Important: If there are ethical or legal restrictions to sharing your data publicly, please explain these restrictions in detail. Please see our guidelines for more information on what we consider unacceptable restrictions to publicly sharing data: http://journals.plos.org/plosone/s/data-availability#loc-unacceptable-data-access-restrictions.

REPLY: We published the minimal data set on Figshare. All code necessary to replicate the analysis is publicly available (DOI: 10.6084/m9.figshare.22013054).

Comment 5. Please review your reference list to ensure that it is complete and correct. If you have cited papers that have been retracted, please include the rationale for doing so in the manuscript text or remove these references and replace them with relevant current references. Any changes to the reference list should be mentioned in the rebuttal letter that accompanies your revised manuscript. If you need to cite a retracted article, indicate the article’s retracted status in the References list and include a citation and full reference for the retraction notice.

REPLY: Done.

Reviewer #1:

The manuscript of Osman et al describes a very well-designed epidemiological study of the etiology of community acquired acute diarrhea in North Lebanon. The work enrolled 360 patients of different ages and included evaluations of clinical outcomes and risk factors. The data obtained is of high interest not only for the Lebanese health system but also for the researchers in the field. The authors found a high frequency of diarrheagenic Escherichia coli pathotypes, specifically EAEC and EPEC, and rotavirus A. The data is adequately presented and analysed, including multivariable logistic regression analyses. A large number of mixed infections was detected and the composition of them is presented. This could be highlighted in the discussion section in the view of the severity or associated clinical symptoms of these case. Also, two other limitations of the study should be considered in the discussion: the wide range of the patients ages (1-91 years old) and the lack of EPEC and EAEC sub-typing (typical and atypical isolates).

Comment 1. page 4, 2nd paragraph: Important studies from the group of Richard Guerrant conducted in Fortaleza, Brazil, should be cited to support the information regarding the impaired cognitive consequences of diarrhea in children leaving in low income areas.

REPLY: Done.

We cited two studies from the group of Dr. Richard Guerrant.

Pinkerton R, Oriá RB, Lima AAM, Rogawski ET, Oriá MOB, Patrick PD, et al. Early Childhood Diarrhea Predicts Cognitive Delays in Later Childhood Independently of Malnutrition. The American journal of tropical medicine and hygiene. 2016;95(5):1004-10. Epub 2016/09/06. doi: 10.4269/ajtmh.16-0150. PubMed PMID: 27601523.

Rogawski McQuade ET, Scharf RJ, Svensen E, Huggins A, Maphula A, Bayo E, et al. Impact of Shigella infections and inflammation early in life on child growth and school-aged cognitive outcomes: Findings from three birth cohorts over eight years. PLOS Neglected Tropical Diseases. 2022;16(9):e0010722. doi: 10.1371/journal.pntd.0010722.

Comment 2. page 5, lines 19-25: The text fragment starting as "For example, 11% of hospitalized...", finishing as "acute gastroenteritis, respectively [31]", would be more appropriate in the Discussion section.

REPLY: We understand the reviewer’s comment. However, we prefer to keep this information in the introduction section to highlight the paucity of data on enteric pathogens circulation in Lebanese communities, which might also be underestimated due to several interlinked factors related to the availability of resources and limitations in the investigation approach.

Comment 3. - page 6: What does WASH stand for?

REPLY: Corrected (water, sanitation and hygiene (WASH)).

Comment 4. Figure 2: Rearrange the bars in descending order of prevalence.

REPLY: Done! We modified figure 2 according to your suggestion.

Comment 5. Figure 3B: Not all types of co-infections detected are represented. This part of Fig. 3 could be presented as Supplementary Material describing the composition (enteric pathogens) found in all mixed infections. This is a relevant result to be presented.

REPLY: We prefer to show the main single and mixed enteric infections and most common pathogen association patterns. As we have more than 140 different patterns, we decided to limit Figure 3B to patterns with ≥5 cases.

As requested by the reviewer, we presented all the patterns in supplementary figure 1.

Comment 6. Detection of co-infections is a common finding in epidemiological studies of the etiology of acute diarrhea in low income countries. However, co-infections of 4 enteric pathogens such as the one presented here (EPEC, EAEC, ETEC and EIEC) is notable, if real. Did the authors investigated the accuracy of such infections in 6 patients? Fecal culture and further colony identification? Separate PCR detection?.

REPLY: Unfortunately, such investigations were not performed due to logistic reasons.

Comment 7. page 14, line 12: Please change the reference 55 for other relevant references to support the point discussed regarding the pathogenesis of EAEC. Several reviews on EAEC pathogenesis would fit better here.

REPLY: Done.

We replaced the reference with two other references as follow:

Ellis SJ, Crossman LC, McGrath CJ, Chattaway MA, Hölken JM, Brett B, et al. Identification and characterisation of enteroaggregative Escherichia coli subtypes associated with human disease. Scientific reports. 2020;10(1):7475. doi: 10.1038/s41598-020-64424-3.

Lääveri T, Vilkman K, Pakkanen SH, Kirveskari J, Kantele A. A prospective study of travellers' diarrhoea: analysis of pathogen findings by destination in various (sub)tropical regions. Clin Microbiol Infect. 2018;24(8):908.e9-.e16. Epub 2017/11/15. doi: 10.1016/j.cmi.2017.10.034. PubMed PMID: 29133155.

Comment 8. page 14, line 16: The authors could discuss here the lack of sensitivity when using serum agglutination detection the so-called classical EPEC O serogroups.

REPLY: Done. We added the requested information as follow.

When tested, slide agglutination assays are performed on E. coli isolates in children younger than two years old and are associated with limited sensitivity and specificity [59].

Nataro JP, Kaper JB. Diarrheagenic Escherichia coli. Clinical Microbiology Reviews. 1998;11(1):142-201. doi: doi:10.1128/CMR.11.1.142. 

Reviewer #2: 

In this work that authors used robust molecular diagnostic methods that target multiple etiologic agents increases the sensitivity and specificity of detection for most enteric pathogens. To better understand the epidemiology of enteric infections, they aimed to determine the prevalence of enteric pathogens, identify risk factors and seasonal variations, and describe associations between pathogens among diarrheic patients in a Lebanese community. The authors found a high prevalence of enteric pathogens, with a predominance of mixed infections. Their statistical models revealed a strong relationship between different diarrheagenic Escherichia coli. Rotavirus A was significantly more likely to occur in the winter season in comparison to the summer. Furthermore, they found that living in rural areas and swimming are considered significant risk factors associated with rotavirus A and norovirus GI/GII infections, respectively. It is a very interesting and well-designed study and the authors’ conclusions are supported by their evidence. I have only minor concerns.

Comment 1. Discussion section, third paragraph: EPEC does not cause dysentery. In fact, the reference 56 does not mention the word “dysentery”.

REPLY: We agree with the reviewer. The respective information has been removed as follows:

For many years, EPEC has been associated with infantile watery diarrhea and dysentery in developing countries [60].

Comment 2. Discussion section, third paragraph, penultimate sentence: rewrite that sentence so as not to give the impression that the authors' work includes bacterial resistance experiments (clearly writing that this is a correlation with reference 57).

REPLY: Done. The information has been clarified as follows:

In our previous retrospective study on bacterial enteric pathogens in children carried out between 2009 and 2015 at the Nini hospital, we described that EPEC was more prevalent than Salmonella spp. and Shigella spp. and showed a higher percentage of resistance to third-generation cephalosporins [61].

Comment 3. The conclusion section is too long and redundant, reduce it and try merging it with the conclusion as the last paragraph (or reduce the conclusion as well, it is redundant).

REPLY: Done. As requested, we reduced the size of the conclusion section. We cut a paragraph of it and transferred it to the discussion section.

---

## [Editor Report · Decision Letter 1]

24 Feb 2023

The indelible toll of enteric pathogens: Prevalence, clinical characterization, and seasonal trends in patients with acute community-acquired diarrhea in disenfranchised communities

PONE-D-22-30455R1

Dear Dr. Hamze,

We’re pleased to inform you that your manuscript has been judged scientifically suitable for publication and will be formally accepted for publication once it meets all outstanding technical requirements.

Kind regards,

Fernando Navarro-Garcia

Academic Editor

PLOS ONE
---

## [Editor Report · Acceptance letter]

1 Mar 2023

PONE-D-22-30455R1 

The indelible toll of enteric pathogens: Prevalence, clinical characterization, and seasonal trends in patients with acute community-acquired diarrhea in disenfranchised communities 

Dear Dr. Hamze:

I'm pleased to inform you that your manuscript has been deemed suitable for publication in PLOS ONE. Congratulations! Your manuscript is now with our production department. 

Kind regards, 

on behalf of

Dr. Fernando Navarro-Garcia 

Academic Editor

PLOS ONE